# AURA: VISUALLY INTERPRETABLE AFFECTIVE UNDERSTANDING VIA ROBUST ARCHETYPES

## ABSTRACT

Text-driven vision–language methods (e.g., CLIP variants) face three persistent hurdles in affective computing: (i) limited support for continuous regression (e.g., Valence–Arousal, VA), (ii) brittle reliance on language prompts, and (iii) the absence of a unified paradigm across expression classification, action unit detection (AUD), and VA regression (VAE). We introduce AURA, a prompt-free framework that projects frozen CLIP visual features into a learnable visual archetype space to perform archetype-grounded predictions and explanations. AURA comprises two components: (1) self-organized archetype discovery, which adaptively allocates the number of archetypes per affective state, assigning denser archetype sets to complex or ambiguous states for fine-grained interpretability, and (2) archetype contextualization, which models interactions among the most relevant archetypes and semantic tokens to enhance structural consistency while suppressing redundancy. Inference reduces to cosine matching between projected features and fixed archetypes. Evaluated across six benchmarks, AURA consistently outperforms prior state-of-the-art while remaining highly efficient. Overall, AURA unifies classification, detection, and regression under a single visual-archetype paradigm, delivering strong accuracy, cognitively aligned interpretability, and excellent efficiency. All models and source code will be released upon acceptance.

## 1 INTRODUCTION

Large-scale vision–language models (VLMs) such as CLIP (Radford et al., 2021) have demonstrated remarkable progress in multimodal representation learning. Benefiting from their paired image–text training paradigm, CLIP's visual embeddings capture more *semantically* rich information compared to other self-supervised visual representation approaches (Jiang et al., 2023; Jose et al., 2025). This advantage has recently been exploited in **interpretable affective understanding**, especially for Facial Expression Recognition (FER) and Action Unit (AU) detection tasks (Zhou et al., 2022b; Li et al., 2024a; Foteinopoulou & Patras, 2024; Chang et al., 2024; Liu et al., 2025). However, existing CLIP-based methods face several critical limitations: **(i) Incompatibility with regression,** Dimensional affective representations, such as the Valence–Arousal (VA) space, are essential for fine-grained affective analysis. CLIP's classification-oriented design, relying on textual descriptions, is fundamentally unsuitable for regression tasks; **(ii) Reliance on brittle linguistic prompts,** Current approaches attempt to describe inherently fuzzy and continuous affective states through elaborately crafted handcrafted or adaptive templates (e.g., CoOp (Zhou et al., 2022b)). FER often demands fine-grained textual descriptions of facial muscle configurations or even multiple sentences to represent a single emotion class (Zhou et al., 2022b; Li et al., 2024a; Foteinopoulou & Patras, 2024). Similarly, AU detection requires translating technical AU codes into approximate linguistic expressions (e.g., "AU5: Upper Lid Raiser" → "wide-open eyes"), a process that is brittle, labor-intensive, and knowledge-dependent (Chang et al., 2024; Liu et al., 2025); **(iii) Lack of a unified framework,** FER and AU detection are typically addressed in isolation with task-specific customization, hindering the development of a generalizable framework that supports classification, detection, and regression in affective computing; and **(iv) Heavy reliance on fine-tuning.** Adapting to affective tasks often requires fine-tuning the visual and/or textual encoders, which increases computational cost and reduces representation generality, confining the model to task-specific settings.

More fundamentally, *text-dependent approaches diverge from how humans naturally interpret emotions*. Cognitive psychology shows that affective understanding is grounded in perceptual similarity

to internalized **visual archetypes**, rather than in **abstract linguistic** reasoning (Rosch, 1975; Fehr & Russell, 1984), motivating a cognitively aligned alternative. We introduce **AURA** (**A**ffective **U**nderstanding via **R**obust **A**rchetypes), a novel *visually interpretable* framework that unifies regression, classification, and detection in a single paradigm. It eliminates reliance on textual descriptions, prompt engineering, and fine-tuning by projecting frozen CLIP visual embeddings into a learnable archetype space to discover an *adaptive* set of *dense, robust* archetypes, offering a perceptually aligned solution.

Notably, our method diverges from existing archetype/clustering/anchor-based approaches (Snell et al., 2017; Deng et al., 2021; Zhou et al., 2023a; Chen et al., 2019), which exhibit three major limitations: *(1) Fixed center numbers.* They predetermine one or a fixed set of archetypes (e.g., Top-$K$) per class, requiring elaborate manual design and leading to brittle representations; *(2) Constrained to classification.* They are predominantly restricted to classification and offer little support for regression tasks; *(3) Lack of fine-grained interpretability.* Their archetypes are typically too coarse to capture subtle affective variations, providing limited semantic interpretability. In contrast, **AURA** introduces three key advances: *(1) Self-organized archetype discovery.* It adaptively induces a dense set of archetypes for each affective state, allocating more archetypes to ambiguous states and fewer to simpler ones, capturing fine-grained variability without manual intervention. *(2) Unified task support.* It unifies regression, classification, and detection in a single framework, overcoming the task-specific tailoring required by prior approaches. *(3) Visually Grounded Semantic Interpretability.* Each sample attains interpretability through its archetypes that are cognitively aligned with human perception. In summary, our contributions include:

- **AURA Framework.** We propose AURA, a novel *visually interpretable* framework that projects frozen CLIP visual embeddings into a learnable archetype space to enable archetype-grounded predictions and explanations, thereby eliminating reliance on textual descriptions, prompt engineering, and encoder fine-tuning, and promoting cognitively aligned modeling.

- **Self-organized Archetype Discovery.** We introduce a self-organized mechanism that adaptively induces an appropriate number of archetypes for each affective state, allocating more to complex or ambiguous emotions and fewer to simpler ones, capturing fine-grained variability.

- **Unified Paradigm and Interpretable Modeling.** AURA establishes a unified framework that consistently supports diverse emotional analysis tasks, including regression, classification, and detection. Critically, it provides cognitively grounded interpretability, ensuring that every model decision is explained through alignment with semantically meaningful archetypes.

- **Effectiveness and Efficiency.** Extensive experiments across six benchmarks demonstrate that AURA consistently achieves state-of-the-art performance with reduced computational cost, underscoring its practicality, efficiency, and generalizability.

## 2 AURA FRAMEWORK

To discover semantically expressive and discriminative visual archetypes, the proposed AURA framework comprises two complementary components, as illustrated in Fig. 1 (Left). **(i) Self-organized Archetype Discovery Module,** discretizes continuous CLIP visual space into a set of semantically rich, compact, and densely distributed archetypes, which serve as the foundation for modeling affective variability. **(ii) Archetype Contextualization Module.** Building upon these archetypes, this module enhances structural coherence and captures nuanced semantic relationships by attending to the most relevant neighbors, thereby promoting contextual alignment.

### 2.1 VISUAL ARCHETYPE-SPACE PROJECTION (VAS).

To embed CLIP visual representations into the archetype space while disentangling affective cues from confounding factors, a *Visual Archetype-Space Projector (VAS)*, $\mathcal{F}(\cdot)$ is introduced. This module compresses high-dimensional embeddings into a compact archetype-centric space and fulfills three functions: (1) steering embeddings toward the archetype space to facilitate effective archetype discovery, (2) attenuating affect-irrelevant variations (e.g., identity, illumination, and background), and (3) amplifying discriminative affective signals under supervised guidance. These functions are critical given the heterogeneity of affective tasks: FER and VA regression require holistic global cues, while AU detection relies on localized details. To capture such nuances, the projector exploits CLIP's

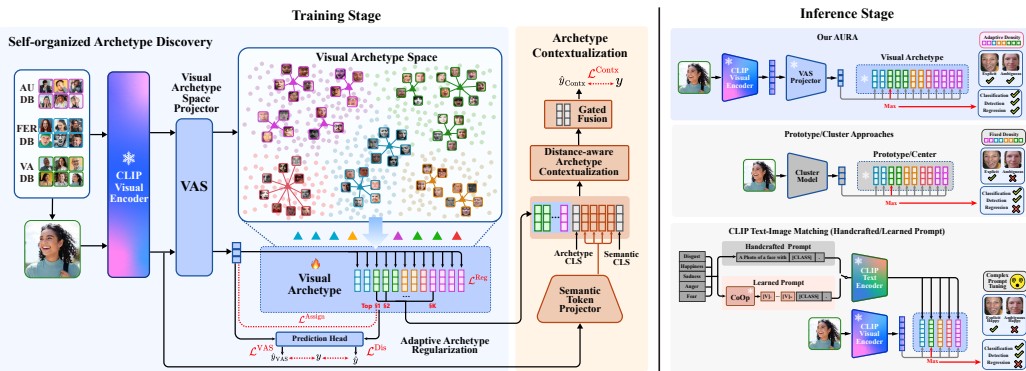

Figure 1: **Left: Training Stage.** Overall architecture of AURA, capable of handling FER, AU detection, and VA regression. AURA consists of two key modules. (1) *Self-organized Archetype Discovery* (blue): frozen CLIP features are projected into a Visual Archetype Space (VAS), followed by sample–archetype assignment with density and expressiveness regularization, which adaptively allocate more archetypes to complex or ambiguous affective states, while representing simpler states with fewer archetypes. (2) *Archetype Contextualization* refines the continuity among archetypes by modeling their contextual dependencies, enhancing semantic consistency and reducing redundancy. **Right: Inference Stage.** Predictions are obtained by cosine matching to fixed archetypes, yielding an efficient and interpretable pipeline. **Comparison.** Existing Prototype/Cluster methods and CLIP text–image matching approaches are also illustrated. Unlike AURA, these cannot perform regression tasks and capture only explicit affective states. Prototype/Cluster methods typically rely on fixed number of centers, whereas AURA adaptively adjusts archetype density. CLIP-based approaches further require prompt engineering, while AURA learns perception-grounded, fine-grained, and interpretable archetypes without manual design.

hierarchical features to derive task-appropriate global- or patch-level embeddings, thereby ensuring the **versatility** of AURA and its capacity to generalize across diverse affective scenarios.

**(i) Global-level Embedding.** Tasks such as FER and VA regression require holistic cues from the entire face. Given an image $x^t$, global embeddings are extracted from the CLIP encoder as $\mathbf{z}_g^t = \mathcal{E}_{\text{CLS}}(x^t) \in \mathbb{R}^D$ and projected into archetype space via a lightweight two-layer projector $\mathbf{f}_t = \mathcal{F}(\mathbf{z}_g^t) \in \mathbb{R}^d$ with $d \ll D$. To attenuate affect-irrelevant variations and enhance discriminative capacity, task-specific supervision is imposed. **For FER**, predictions are obtained by a linear classifier $\phi_{\text{Exp}}(\cdot)$, optimized with cross-entropy, with weight $w_y = \frac{N}{y \cdot n_y}$ compensates for class imbalance:

$$\mathcal{L}_{\text{Exp}}^{\text{VAS}} = -w_y \log \phi_{\text{Exp}}(\mathbf{f}_{\text{Exp}})_y. \tag{1}$$

**For VA regression**, regression heads $\phi_V(\cdot)$ and $\phi_A(\cdot)$ operate on valence and arousal features, and the objective integrates concordance correlation (CCC) with mean squared error (MSE):

$$\mathcal{L}_{\text{VA}}^{\text{VAS}} = \tfrac{1}{2}\big[\text{CCC}(\phi_V(\mathbf{f}_V), y_V) + \text{CCC}(\phi_A(\mathbf{f}_A), y_A)\big] + \alpha_{\text{mse}} \tfrac{1}{2}\big[\text{MSE}(\phi_V(\mathbf{f}_V), y_V) + \text{MSE}(\phi_A(\mathbf{f}_A), y_A)\big]. \tag{2}$$

**(ii) Patch-level Embedding.** AU detection targets localized muscle activations and therefore requires fine-grained patch features. Patch embeddings $\mathbf{z}_p^{\text{AU}} \in \mathbb{R}^{L \times D}$ are extracted from the final hidden layer of CLIP, where $L$ denotes the number of patches. A lightweight Transformer $\mathcal{T}_{\text{proj}}(\cdot)$ augments them with a learnable $[\text{CLS}]$ token and positional encodings, yielding contextualized features $\tilde{\mathbf{z}}^{\text{AU}} = \mathcal{T}_{\text{proj}}(\mathbf{z}_p^{\text{AU}}) \in \mathbb{R}^D$. For each AU $m$, an independent projector $\mathcal{F}_{(m)}(\cdot)$ maps $\tilde{\mathbf{z}}^{\text{AU}}$ into archetype space, followed by a detection head $\phi_{(m)}(\cdot)$, optimized by binary cross-entropy across all $M$ AUs:

$$\mathcal{L}_{\text{AU}}^{\text{VAS}} = \tfrac{1}{M} \sum_{m=1}^{M} \text{BCE}\big(y_m, \phi_{(m)}(\mathcal{F}_{(m)}(\tilde{\mathbf{z}}^{\text{AU}}))\big). \tag{3}$$

## 2.2 SELF-ORGANIZED ARCHETYPE DISCOVERY MODULE

To discover self-organized, perception-grounded & task-adaptive visual archetypes, a *Self-organized Archetype Discovery* module is introduced. It operates via two complementary stages: *(i) Discriminative Archetype Assignment*, which aligns projected visual features with archetypes under label supervision, strengthening their discriminative capacity; and *(ii) Adaptive Archetype Regularization*, which self-organizes both the density and distribution of archetypes, refining their semantic fidelity and compactness. The whole architecture is illustrated in left (blue) part of Fig. 1.

### 2.2.1 DISCRIMINATIVE ARCHETYPE ASSIGNMENT

This step enforces discriminative alignment between samples and archetypes, ensuring that each sample is mapped to the most semantically representative archetype in the latent space. Formally, let $\mathcal{A} = \{\mathbf{a}^1, \ldots, \mathbf{a}^K\} \subset \mathbb{R}^d$ denote a learnable set of $K$ archetypes, where each $\mathbf{a}^k$ serves as a semantically meaningful anchor in the latent space. Given a projected feature $\mathbf{f} \in \mathbb{R}^d$, an assignment operator $\mathcal{Q}(\cdot; \mathcal{A})$ maps $\mathbf{f}$ to the most semantically aligned archetype under cosine similarity. To guarantee scale-invariant matching, both features and archetypes are $\ell_2$-normalized. The resulting assigned archetype is defined as

$$\mathbf{a}^{k^*} = \mathcal{Q}(\mathbf{f}; \mathcal{A}), \quad k^* = \arg\max_k \left\langle \frac{\mathbf{f}}{\|\mathbf{f}\|_2}, \frac{\mathbf{a}^k}{\|\mathbf{a}^k\|_2} \right\rangle. \tag{4}$$

To enable semantically faithful archetype representations, two complementary objectives are formulated: (i) an *Archetype Assignment Loss* that stabilizes learning and enforces consistent alignment, and (ii) a *Task-aware Supervision Loss* that preserves semantic fidelity with respect to affective tasks.

**(i) Archetype Assignment Loss ($\mathcal{L}^{\text{Assign}}$).** To ensure stable archetype discovery and robust feature discretization, a dual-term objective is defined for each projected feature $\mathbf{f}$ and its assigned archetype $\mathbf{a}$. The first term adapts the archetype $\mathbf{a}$ toward the encoder distribution, while the second enforces a commitment constraint from $\mathbf{f}$ to $\mathbf{a}$, preventing feature drift and promoting stable alignment. Formally,

$$\mathcal{L}^{\text{Assign}} = \underbrace{\left\| \text{sg}[\mathbf{f}] - \mathbf{a} \right\|_2^2}_{\text{archetype adaptation}} + \beta \cdot \underbrace{\left\| \mathbf{f} - \text{sg}[\mathbf{a}] \right\|_2^2}_{\text{commitment penalty}}, \tag{5}$$

where $\text{sg}[\cdot]$ denotes the stop-gradient operator and $\beta$ balances adaptation and commitment. This symmetric formulation allows archetypes to self-organize around the latent feature distribution, while ensuring that encoder outputs consistently conform to the discovered archetype structure.

**(ii) Archetype Discriminative Loss ($\mathcal{L}^{\text{Dis}}$).** To ensure archetypes remain both task-discriminative and perceptually grounded, the supervision objective is coupled with the archetype assignment process. This joint formulation aligns archetype learning with human-consistent perception while enabling adaptive specialization to the heterogeneous label of FER, AU detection, and VA regression.

*For FER.* A shared archetype set is defined across all expression classes, denoted as $\mathcal{A}_{\text{Exp}} = \{\mathbf{a}_{\text{Exp}}^1, \ldots, \mathbf{a}_{\text{Exp}}^{K_{\text{Exp}}}\}$, where $K_{\text{Exp}}$ is the number of archetypes. Given a projected feature $\mathbf{f}$, archetype assignment yields $\mathbf{a}^k = \mathcal{Q}(\mathbf{f}; \mathcal{A}_{\text{Exp}})$. The expression class associated with the selected archetype is then predicted as $\hat{y}_{\mathbf{a}^k} = \phi_{\text{Exp}}(\mathbf{a}^k)$, where $\phi_{\text{Exp}}(\cdot)$ denotes the shared classifier in Eq. 1. To enforce semantic alignment, a weighted cross-entropy loss $\mathcal{L}_{\text{Exp}}^{\text{Dis}}(\hat{y}_{\mathbf{a}^k}, y)$ is employed, consistent with Eq. 1.

*For VA Regression.* Two distinct archetype sets are defined, $\mathcal{A}_{\text{V}} = \{\mathbf{a}_{\text{V}}^1, \ldots, \mathbf{a}_{\text{V}}^{K_{\text{V}}}\}$ and $\mathcal{A}_{\text{A}} = \{\mathbf{a}_{\text{A}}^1, \ldots, \mathbf{a}_{\text{A}}^{K_{\text{A}}}\}$, corresponding to valence and arousal. Projected features $\mathbf{f}_{\text{V}}$ and $\mathbf{f}_{\text{A}}$ are assigned with archetypes via quantization: $\mathbf{a}_{\text{V}}^k = \mathcal{Q}(\mathbf{f}_{\text{V}}; \mathcal{A}_{\text{V}})$ and $\mathbf{a}_{\text{A}}^k = \mathcal{Q}(\mathbf{f}_{\text{A}}; \mathcal{A}_{\text{A}})$. Predictions are then obtained as $\hat{y}_{\text{V}} = \phi_{\text{V}}(\mathbf{a}_{\text{V}}^k)$ and $\hat{y}_{\text{A}} = \phi_{\text{A}}(\mathbf{a}_{\text{A}}^k)$, where $\phi_{\text{V}}(\cdot)$ and $\phi_{\text{A}}(\cdot)$ are shared regression heads in Eq. 2. For each dimension $t \in \{\text{V}, \text{A}\}$, supervision is imposed through $\mathcal{L}_t^{\text{Dis}}$, consistent with Eq. 2.

*For AU Detection.* A distinct archetype set $\mathcal{A}_{(m)} = \{\mathbf{a}_{(m)}^1, \ldots, \mathbf{a}_{(m)}^{K_{(m)}}\}$ is defined for each AU $m$. The projected feature $\mathbf{f}_{(m)}$ is independently assigned with an archetype via $\mathbf{a}_{(m)}^k = \mathcal{Q}(\mathbf{f}_{(m)}; \mathcal{A}_{(m)})$. The activation score is then obtained as $\hat{y}_{(m)} = \phi_{(m)}(\mathbf{a}_{(m)}^k)$, where $\phi_{(m)}(\cdot)$ shared in Eq. 3. Each AU archetype set is supervised with $\mathcal{L}_{(m)}^{\text{Dis}}$, consistent with Eq. 3.

### 2.2.2 ADAPTIVE ARCHETYPE REGULARIZATION

To enhance the quality of learned archetypes while accommodating task-specific characteristics, AURA introduces adaptive regularizations focusing on two key aspects: *density* and *expressiveness*. **Density** governs the allocation of archetypes per affective state, ensuring that simple states are represented by fewer archetypes, while ambiguous or highly variable states receive denser coverage. **Expressiveness** captures the semantic fidelity of each archetype, requiring compactness within

homogeneous states and discriminativeness across heterogeneous ones. By jointly regularizing them, AURA self-organizes archetypes into a semantically faithful and task-adaptive structure.

**For FER Task.** With discrete emotion labels, a complementary regularization strategy is devised to enhance both the robustness and stability of learned archetypes by enforcing *(i) intra-class compactness*, *(ii) inter-class separation*, and *(iii) archetype diversity*. Through their combined effect, the archetype space self-organizes into a stable configuration, where **each emotion class is allocated an appropriate number of archetypes** that faithfully reflects its intrinsic variability. Formally, Each archetype $\mathbf{a}^k$ is assigned to an emotion class via $c_k = \arg\max_u [\phi_{\mathrm{Exp}}(\mathbf{a}^k)]_u$, where $\phi_{\mathrm{Exp}}(\cdot)$ is the classifier in Eq. 1, and its confidence is measured by $w^k = \max_u [\mathrm{softmax}(\phi_{\mathrm{Exp}}(\mathbf{a}^k))]_u$. For each class $c$, the associated archetype set is $\mathcal{A}_c = \{k \mid c_k = c\}$ with class center $\boldsymbol{\mu}_c = \frac{1}{|\mathcal{A}_c|} \sum_{k \in \mathcal{A}_c} \mathbf{a}^k$, normalized as $\tilde{\boldsymbol{\mu}}_c = \boldsymbol{\mu}_c / \|\boldsymbol{\mu}_c\|_2$. The three optimization objectives are defined as:

*(i) Intra-class Compactness Loss ($\mathcal{L}_{intra}$):* encourages archetypes to align with their respective class centers, with uncertain archetypes (low $w^k$) penalized more heavily:

$$\mathcal{L}_{\mathrm{intra}} = \tfrac{1}{K} \sum_{k=1}^{K} (1 - w^k) \cdot \big(1 - \cos(\tilde{\mathbf{a}}^k, \tilde{\boldsymbol{\mu}}_{c_k})\big). \tag{6}$$

*(ii) Inter-class Separation Loss ($\mathcal{L}_{inter}$):* enforces distinctiveness between classes by penalizing pairs of centers whose similarity exceeds a margin $m$:

$$\mathcal{L}_{\mathrm{inter}} = \tfrac{1}{|\mathcal{S}|} \sum_{(c,c') \in \mathcal{S}} \max\big(0, \cos(\tilde{\boldsymbol{\mu}}_c, \tilde{\boldsymbol{\mu}}_{c'}) - m\big). \tag{7}$$

*(iii) Archetype Diversity Loss ($\mathcal{L}_{div}$):* encourages semantic dispersion while preventing collapse by balancing spread from the global mean and variance across classes. Let $\bar{\boldsymbol{\mu}} = \frac{1}{C} \sum_{c=1}^{C} \tilde{\boldsymbol{\mu}}_c$ be the global mean, $d_c = \|\tilde{\boldsymbol{\mu}}_c - \bar{\boldsymbol{\mu}}\|_2$, and $\bar{d} = \frac{1}{C} \sum_{c=1}^{C} d_c$. The loss is:

$$\mathcal{L}_{\mathrm{div}} = -\tfrac{1}{C} \sum_{c=1}^{C} d_c - \tfrac{1}{C} \sum_{c=1}^{C} (d_c - \bar{d})^2. \tag{8}$$

The total FER archetype regularization loss is then given by $\mathcal{L}^{\mathrm{Reg}} = \mathcal{L}_{\mathrm{intra}} + \mathcal{L}_{\mathrm{inter}} + \mathcal{L}_{\mathrm{div}}$.

**For AU Detection and VA Regression.** Unlike FER, these tasks lack reliable discrete class boundaries. VA regression is inherently continuous, while AU detection, although binary in label space, involves subtle intensity variations that cannot be faithfully captured by strict binary modeling. To address this, AU predictions are treated as continuous activation scores within $[0, 1]$. Accordingly, a class-free archetype regularization strategy is adopted, which jointly enforces *(i) Score-aware Attraction Loss*, *(ii) Score-aware Repulsion Loss*, and *(iii) archetype diversity*. This design ensures that archetypes remain semantically expressive and robust, even without explicit class supervision.

*(i) Score-aware Attraction Loss ($\mathcal{L}_{\mathrm{attr}}$):* encourages archetypes with similar predicted scores to cluster in the feature space. Let $\mathbf{a}^i$ and $\mathbf{a}^j$ denote two archetypes with predicted scores $\hat{y}^i = \phi_{\mathrm{task}}(\mathbf{a}^i)$ and $\hat{y}^j = \phi_{\mathrm{task}}(\mathbf{a}^j)$, where $\phi_{\mathrm{task}}(\cdot)$ is a shared regression head (sigmoid-activated for AU). Define the score gap $\Delta^{ij} = |\hat{y}^i - \hat{y}^j|$ and cosine distance $d^{ij} = \frac{1 - \cos(\mathbf{a}^i, \mathbf{a}^j)}{2}$. The candidate set $\mathcal{S}_{\mathrm{attr}} = \{(i, j) \mid \Delta^{ij} < m\}$ contains archetype pairs with similar scores, where $m$ is a soft margin. The loss is then:

$$\mathcal{L}_{\mathrm{attr}} = \frac{1}{|\mathcal{S}_{\mathrm{attr}}|} \sum_{(i,j) \in \mathcal{S}_{\mathrm{attr}}} \big[\max(0, d^{ij} - m)\big]^2, \tag{9}$$

which penalizes unnecessary separation among archetypes aligned with similar activation levels.

*(ii) Score-aware Repulsion Loss ($\mathcal{L}_{\mathrm{repul}}$):* This term enforces separation between archetypes associated with dissimilar predicted scores. Let $\Delta^{ij} = |\hat{y}^i - \hat{y}^j|$ and $d^{ij} = \frac{1 - \cos(\mathbf{a}^i, \mathbf{a}^j)}{2}$ denote the score gap and cosine distance between two archetypes, respectively. The candidate set $\mathcal{S}_{\mathrm{repul}} = \{(i, j) \mid \Delta^{ij} \geq m\}$ collects pairs with divergent scores, where $m$ is a soft margin. The repulsion loss is then defined as

$$\mathcal{L}_{\mathrm{repul}} = \frac{1}{|\mathcal{S}_{\mathrm{repul}}|} \sum_{(i,j) \in \mathcal{S}_{\mathrm{repul}}} \big[\max(0, m - d^{ij})\big]^2, \tag{10}$$

which prevents collapse by pushing apart archetypes encoding distinct affective states.

*(iii) Archetype Diversity* ($\mathcal{L}_{\text{div}}$): To avoid collapse and promote semantic dispersion, archetypes are further regularized with the diversity objective $\mathcal{L}_{\text{div}}$ as defined in Eq. 8. The overall regularization loss for AU detection and VA Regression defined as: $\mathcal{L}^{\text{Reg}} = \mathcal{L}_{\text{attr}} + \mathcal{L}_{\text{repul}} + \mathcal{L}_{\text{div}}$.

### 2.2.3 TOTAL SELF-ORGANIZED ARCHETYPE DISCOVERY LOSS

Finally, we combine the Discriminative Archetype Assignment Losses and the Adaptive Archetype Regularization Losses to define the Self-organized Archetype Discovery Loss for all tasks as:

$$\mathcal{L}^{\text{Arc}} = \mathcal{L}^{\text{Assign}} + \mathcal{L}^{\text{Dis}} + \mathcal{L}^{\text{Reg}}. \tag{11}$$

### 2.3 ARCHETYPE CONTEXTUALIZATION MODULE

The Self-organized Archetype Discovery module learns archetypes independently, but such isolation overlooks semantic continuity among those jointly relevant to a sample. To address this, an **Archetype Contextualization (AC)** module (Fig. 1, left brown part) enforces structural consistency across related archetypes. For each input, the top-$K$ similar archetypes and the projected semantic token are processed by a lightweight Transformer encoder to capture contextual dependencies. This adapts archetypes to their semantic neighbors, enhancing consistency and reducing redundancy.

#### 2.3.1 SEMANTIC TOKEN PROJECTOR.

Archetype Contextualization aims to capture context-dependent relations among archetypes. For each sample, in addition to retrieving its top-$K$ most relevant archetypes, the semantic representation obtained via the Semantic Token Projector is also incorporated. Same with Sec. 2.1, global embeddings used for FER and VA regression, while patch embeddings are adopted for AU detection.

**(i) Global Semantic Token Projector.** An attention-based projection module is introduced to derive a compact set of $N_{\text{tok}}$ semantic tokens from global CLIP embeddings. Given $\mathbf{z}_{\text{g}}^{\text{t}}$, a learnable query set $\mathbf{S} = [\mathbf{s}^1; \ldots; \mathbf{s}^{N_{\text{tok}}}] \in \mathbb{R}^{N_{\text{tok}} \times d_{\text{tok}}}$ is defined, each $\mathbf{s}^n$ is intended to capture a distinct affective attribute. For token index $n$, token-specific key and value projections $f_K^{(n)}, f_V^{(n)} : \mathbb{R}^d \to \mathbb{R}^{d_{\text{tok}}}$ generate

$$\mathbf{k}^{(n)} = f_K^{(n)}(\mathbf{z}_{\text{g}}^{\text{t}}), \quad \mathbf{v}^{(n)} = f_V^{(n)}(\mathbf{z}_{\text{g}}^{\text{t}}). \tag{12}$$

Each query $\mathbf{s}^n$ attends to its key-value pair through a multi-head attention operator $\Psi(\cdot)$, yielding

$$\mathbf{t}^n = \Psi(\mathbf{s}^n, \mathbf{k}^{(n)}, \mathbf{v}^{(n)}). \tag{13}$$

The tokens $\{\mathbf{t}^n\}_{n=1}^{N_{\text{tok}}}$ are aggregated and normalized into global semantic token $\mathbf{T}^{\text{g}} \in \mathbb{R}^{N_{\text{tok}} \times d_{\text{tok}}}$.

**(ii) Patch Semantic Token Projector.** To capture fine-grained structural cues essential for AU, we exploit patch embeddings $\mathbf{z}_{\text{p}}^{\text{AU}} \in \mathbb{R}^{L \times D}$. A Transformer $\mathcal{T}_{\text{tok}}(\cdot)$ aggregates local context, where $N_{\text{tok}}$ learnable tokens $\{\mathbf{s}^n\}_{n=1}^{N_{\text{tok}}}$ are prepended to the patch sequence:

$$[\tilde{\mathbf{t}}^1; \ldots; \tilde{\mathbf{t}}^{N_{\text{tok}}}] = \mathcal{T}_{\text{tok}}([\mathbf{s}^1; \ldots; \mathbf{s}^{N_{\text{tok}}}] \, \| \, \mathbf{z}_{\text{p}}^{\text{AU}}), \quad \tilde{\mathbf{t}}^n \in \mathbb{R}^D.$$

Each $\tilde{\mathbf{t}}^n$ is projected through an AU-specific linear layer for each AU to yield semantic tokens $\mathbf{t}^n \in \mathbb{R}^{d_{\text{tok}}}$, forming the patch-aware semantic token $\mathbf{T}^{\text{p}} = [\mathbf{t}^1; \ldots; \mathbf{t}^{N_{\text{tok}}}] \in \mathbb{R}^{N_{\text{tok}} \times d_{\text{tok}}}$.

#### 2.3.2 DISTANCE-AWARE ARCHETYPE CONTEXTUALIZATION

To capture contextual dependencies among semantically related archetypes and mitigate redundancy, we retrieve the top-$K$ nearest archetypes for each sample. The selected archetypes $\{\mathbf{a}^{(1)}, \ldots, \mathbf{a}^{(K)}\}$ are then transformed into **distance-aware** semantic tokens:

$$\hat{\mathbf{a}}^{(k)} = \mathbf{W}_{\text{proj}} \mathbf{a}^{(k)} + \mathbf{b}_{\text{proj}} + \mathbf{W}_{\text{dist}} \, d_k + \mathbf{b}_{\text{dist}}, \tag{14}$$

where $\mathbf{a}^{(k)} \in \mathbb{R}^d$ denotes the $k$-th nearest archetype, $d_k$ is its cosine distance to the projected sample feature $\mathbf{f}$, $\mathbf{W}_{\text{proj}} \in \mathbb{R}^{d_{\text{tok}} \times d}$ and $\mathbf{W}_{\text{dist}} \in \mathbb{R}^{d_{\text{tok}} \times 1}$ are learnable projection matrices, and $\mathbf{b}_{\text{proj}}, \mathbf{b}_{\text{dist}} \in \mathbb{R}^{d_{\text{tok}}}$ are bias terms. This yields the sequence of semantic tokens $\hat{\mathbf{A}} = [\hat{\mathbf{a}}^{(1)}; \ldots; \hat{\mathbf{a}}^{(K)}] \in \mathbb{R}^{K \times d_{\text{tok}}}$.

Let $\mathbf{T} = [\mathbf{t}^1; \ldots; \mathbf{t}^{N_{\text{tok}}}] \in \mathbb{R}^{N_{\text{tok}} \times d_{\text{tok}}}$ denote the semantic tokens obtained in Section 2.3.1. Two special tokens, a semantic CLS token $\mathbf{c}_{\text{sem}}$ and an archetype CLS token $\mathbf{c}_{\text{arc}}$, are prepended:

$$\mathbf{X}_{\text{Contx}} = [\mathbf{c}_{\text{sem}}; \ \mathbf{T}; \ \mathbf{c}_{\text{arc}}; \ \hat{\mathbf{A}}] \in \mathbb{R}^{(2 + N_{\text{tok}} + K) \times d_{\text{tok}}}. \tag{15}$$

After adding positional encodings, the sequence is Contxd by a Transformer encoder $\mathcal{T}_{\text{Contx}}(\cdot)$:

$$\tilde{\mathbf{X}} = \mathcal{T}_{\text{Contx}}(\mathbf{X}_{\text{Contx}}). \tag{16}$$

We then extract the updated tokens $\tilde{\mathbf{c}}_{\text{sem}}$ and $\tilde{\mathbf{c}}_{\text{arc}}$, and concatenate them into $\mathbf{u} = [\tilde{\mathbf{c}}_{\text{sem}}; \ \tilde{\mathbf{c}}_{\text{arc}}] \in \mathbb{R}^{2d_{\text{tok}}}$. A softmax-based gating module $\mathcal{G}(\cdot) : \mathbb{R}^{2d_{\text{tok}}} \to \mathbb{R}^2$ generates the fusion weights $\boldsymbol{\alpha} = \mathcal{G}(\mathbf{u}) = [\alpha_{\text{sem}}, \ \alpha_{\text{arc}}]$, and the fused representation is computed as $\mathbf{f}_{\text{fused}} = \alpha_{\text{sem}} \cdot \tilde{\mathbf{c}}_{\text{sem}} + \alpha_{\text{arc}} \cdot \tilde{\mathbf{c}}_{\text{arc}}$. Finally, $\mathbf{f}_{\text{fused}}$ is normalized and fed into a prediction head $\phi_{\text{task}}(\cdot)$ to yield $\hat{y}_{\text{Contx}}$. The objective is defined as:

$$\mathcal{L}^{\text{Contx}} = \mathcal{L}_{\text{task}}(\hat{y}_{\text{Contx}}, y), \tag{17}$$

where $\mathcal{L}_{\text{task}}$ corresponds to weighted cross-entropy loss for FER, binary cross-entropy loss for AU detection, and concordance correlation with mean squared error loss for VA regression.

# 3 TRAINING OBJECTIVE AND INFERENCE STRATEGY

**Training Objective.** The learning process is guided by a composite loss comprising three components: (i) $\mathcal{L}^{\text{VAS}}$ for visual archetype-space projection, (ii) $\mathcal{L}^{\text{Arc}}$ for self-organized archetype discovery, and (iii) $\mathcal{L}^{\text{Contx}}$ for archetype contextualization. The overall optimization objective is defined as

$$\mathcal{L}_{\text{total}} = \lambda_{\text{Proj}} \cdot \mathcal{L}^{\text{VAS}} + \lambda_{\text{Arc}} \cdot \mathcal{L}^{\text{Arc}} + \lambda_{\text{Contx}} \cdot \mathcal{L}^{\text{Contx}}, \tag{18}$$

where $\lambda_{\text{Proj}}, \lambda_{\text{Arc}}, \lambda_{\text{Contx}}$ are non-negative weighting coefficients.

**Adaptive Archetype Number.** AURA initially set a relatively large $K_{\text{max}}$ to provide sufficient capacity, and during training the model—through the Adaptive Archetype Regularization and the Archetype Contextualization Module—automatically converges to a much smaller and stable number of active archetypes $K_{\text{stable}} \ll K_{\text{max}}$. At inference, only these active archetypes are retained while unused ones are discarded. More details are provided in Appendix E.

**Inference Strategy.** During inference, the well learned archetype set is fixed and the contextualization module is omitted. A CLIP visual embedding $\mathbf{z}$ is first projected into archetype space as $\mathbf{f} = \mathcal{F}(\mathbf{z})$, after which $\mathbf{f}$ is matched to its nearest archetype $\hat{\mathbf{a}} = \mathcal{Q}(\mathbf{f}; \mathcal{A})$ using cosine similarity. The task-specific head then generates the final prediction as $\hat{y} = \phi_{\text{task}}(\hat{\mathbf{a}})$. This lightweight and interpretable pipeline relies *solely* on the discovered archetypes and the projection module.

# 4 EXPERIMENTS

We evaluate the effectiveness of AURA on both **image** and **video** benchmarks. For FER, experiments are conducted on large-scale in-the-wild datasets AffectNet-7/8 and RAF-DB. For AU detection, evaluations are performed on the in-the-wild image dataset EmotioNet and further assessed at the video level on DISFA. For VA regression, we use AffectNet-VA. Additional experimental details, including implementation, datasets, evaluation protocols, and extended archetype visualizations and detailed analysis, are provided in the supplementary material (Appendix A and Appendix C).

## 4.1 COMPARISON WITH SOTA & CLIP VARIANTS

Across the three tasks summarized in Table 1, AURA delivers consistently superior performance: for FER, it achieves the best results on all three datasets, evidencing strong cross-dataset generalization; for AUD, it attains the highest average F1 on both the image (EmotioNet) and video (DISFA) settings, surpassing prior state of the art; and for VAE, it yields the highest average CCC as well as the best per-dimension CCC for Valence and Arousal. Notably, despite leveraging discrete archetypes, AURA effectively supports continuous regression, preserving fine-grained affective nuances. Moreover, when compared against CLIP-based approaches under identical protocols, AURA maintains clear advantages: in VAE, both *CLIP-FT* (fine-tuning with discretized prompts) and prompt-learning variants (*CoOp/CoCoOp*) obtain low CCC; in FER and AUD, CLIP variants fine-tuned with class-level prompts or learnable textual prompt consistently trail task-specific methods, with *CoOp/CoCoOp*

Table 1: Comparison of AURA with SOTA methods. Results are reported in F1-score (%) for AU detection, accuracy for FER, and CCC for VA regression. Statistical significance is verified by paired $t$-tests ($p < 0.05$).

**Facial Expression Recognition (Accuracy in %)**

| Model | RAF-DB ↑ | AffNet-7 ↑ | AffNet-8 ↑ | Params ↓ | FLOPs ↓ |
|---|---|---|---|---|---|
| MRAN (Chen et al., 2023) | 90.03 | 66.31 | 62.48 | 60.52 | 3.89 |
| AMP-Net (Liu et al., 2022) | 89.25 | 64.54 | 61.74 | 105.67 | 4.73 |
| MA-Net (Zhao et al., 2021) | 88.42 | 64.53 | 60.29 | 63.50 | 3.65 |
| DACL (Farzaneh & Qi, 2021) | 87.78 | 65.20 | – | 103.04 | 1.92 |
| EAC (Zhang et al., 2022) | 89.99 | 65.32 | 62.13 | 29.50 | 10.30 |
| DR-FER (Li et al., 2023) | 91.61 | 67.54 | 63.60 | 48.20 | 13.20 |
| TransFER (Xue et al., 2021) | 90.91 | 66.23 | – | 65.20 | 15.30 |
| VTFF (Ma et al., 2021) | 88.14 | 64.80 | 61.85 | 51.80 | 6.08 |
| FRA (Gao & Patras, 2024) | 90.76 | 65.85 | 62.55 | 24.00 | – |
| S2D (Chen et al., 2024) | 92.57 | 67.62 | 63.08 | 9.00 | – |
| FER-VMamba (Ma et al., 2025) | 92.37 | 67.34 | 64.55 | 4.12 | 0.58 |
| TriBAN (Kim & Choi, 2025) | 91.43 | 66.05 | 62.49 | 67.38 | 4.81 |
| POSTER++ (Mao et al., 2025) | 92.21 | 67.49 | 63.77 | 43.70 | 8.40 |
| UA-FER (Zhou et al., 2025) | 92.59 | 66.95 | 62.42 | 28.74 | 11.27 |
| MHAN (Wang et al., 2025) | 92.57 | 67.74 | 65.08 | 4.27 | 0.55 |
| CLIPER (Li et al., 2024a) | 91.61 | 66.29 | 61.98 | 149.25 | 103.7 |
| CLIP-FT (Radford et al., 2021) | 90.38 | 64.76 | 60.21 | 149.25 | 21.8 |
| CoOp (Zhou et al., 2022b) | 84.65 | 61.25 | 57.76 | 85.71 | 4.26 |
| CoCoOp (Zhou et al., 2022a) | 85.89 | 63.14 | 59.66 | 86.80 | 4.35 |
| **AURA (Ours)** | **94.04 ± 0.17** | **68.43 ± 0.15** | **65.16 ± 0.24** | **3.30** | **0.26** |

**Action Unit Detection (F1 Score in %)**

| Methods | DISFA | EmotioNet |
|---|---|---|
| KSRL (Chang & Wang, 2022) | 64.5 | - |
| CTC (Zhou et al., 2023b) | 57.3 | 64.4 |
| FBNet (Kollias et al., 2019) | 62.0 | 54.0 |
| AUNet (Romero et al., 2022) | 59.7 | 64.6 |
| MEGraphAU (Luo et al., 2022) | 62.4 | 64.9 |
| SITU (Liu et al., 2023) | 62.9 | 64.2 |
| CLEF (Zhang et al., 2023b) | 64.8 | - |
| FUXI (Zhang et al., 2023a) | 63.3 | 65.4 |
| MCM (Zhang et al., 2024) | 64.3 | - |
| EmoLA (Li et al., 2024b) | 65.1 | - |
| MDHRM (Wang et al., 2024) | 66.2 | - |
| CLIPER (Li et al., 2024a) | 54.9 | 56.4 |
| CLIP-FT (Radford et al., 2021) | 60.2 | 61.9 |
| CoOp (Zhou et al., 2022b) | 58.4 | 59.7 |
| CoCoOp (Zhou et al., 2022a) | 59.9 | 61.6 |
| **AURA (Ours)** | **66.9 ± 0.3** | **67.3 ± 0.4** |

**Valence-Arousal Estimation (CCC in %)**

| Model | CCC-V | CCC-A | CCC-VA |
|---|---|---|---|
| VA-StarGAN (Kollias, 2020) | 61.0 | 48.0 | 54.5 |
| FaceBehaviorNet (Kollias, 2021) | 66.0 | 58.0 | 62.0 |
| Emotion-GCN (Antoniadis, 2021) | 76.7 | 64.9 | 70.8 |
| EMOCA (Daněček, 2022) | 77.0 | 68.0 | 72.5 |
| EmoNet (Toisoul et al., 2021) | 73.0 | 65.0 | 69.0 |
| BFsGP (Fu et al., 2025) | 72.4 | 64.5 | 68.5 |
| CAGE (Wagner et al., 2024) | 71.6 | 64.2 | 67.9 |
| Ig3D (Dong et al., 2024) | 72.4 | 65.0 | 68.7 |
| CMFR (He & Da, 2025) | 77.0 | 69.0 | 73.0 |
| CLIP-FT (Radford et al., 2021) | 57.4 | 43.3 | 50.3 |
| CoOp (Zhou et al., 2022a) | 51.2 | 37.7 | 44.5 |
| CoCoOp (Zhou et al., 2022b) | 52.5 | 38.3 | 45.4 |
| **AURA (Ours)** | **78.0 ± 0.4** | **70.2 ± 0.1** | **74.1 ± 0.3** |

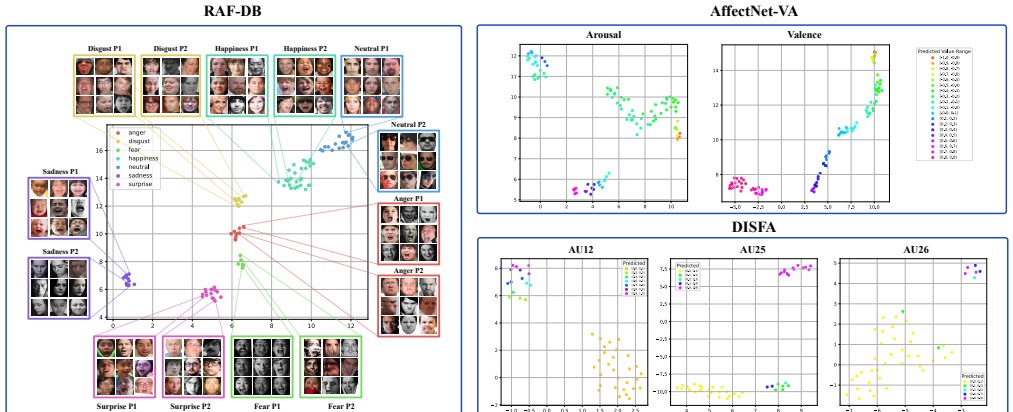

Figure 2: **Left**: visualization of learned archetype distributions on RAF-DB and their assigned samples, demonstrating the interpretability of AURA archetypes. **Right top**: archetypes on AffectNet-VA, illustrating fine-grained affective perception and their effectiveness for regression tasks. **Right bottom**: archetypes on DISFA for AU detection, showing clear distinction between active and inactive AUs.

further degraded by class imbalance and subtle AU activations. While *CLIPER* is competitive on FER, it fares poorly on AUD and VAE and incurs substantial overhead due to its multi-prompt design.

**Training Efficiency** As shown in Table 1, We compare training parameters and FLOPs of AURA with SOTA. AURA's key advantage lies in building on a pretrained CLIP visual encoder, allowing all features to be pre-extracted and eliminating the need for repeated inference during training or evaluation. AURA achieves superior performance with the lowest parameter count and FLOPs. At inference, AURA relies solely on cosine similarity with the learned archetypes, incurring negligible computational overhead.

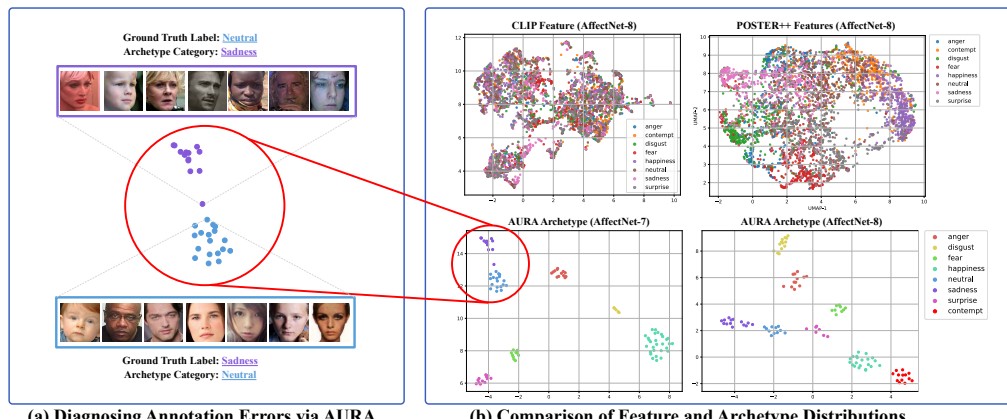

(a) Diagnosing Annotation Errors via AURA     (b) Comparison of Feature and Archetype Distributions

Figure 3: **(a)** Annotation error diagnosis with AURA, demonstrating its ability to detect mislabeled samples and provide fine-grained interpretability; **(b)** UMAP visualization of archetypes, original CLIP features, and POSTER++ features on AffectNet-7/8.

## 4.2 ARCHETYPES SEMANTIC AND DISTRIBUTION ANALYSIS

**Visual Perception Grounded**: Fig. 2 (left) shows that AURA's archetypes grouping visually and semantically coherent expressions without relying on textual prompts. Importantly, the archetypes align not only across different emotion categories but also across varying intensity, consistently matching samples with highly similar visual cues. By capturing both subtle variations and dominant modes, the archetypes reflect perceptually grounded structures that bridge low-level cues and high-level affective semantics, making the affective space interpretable, cognitively consistent. **Diagnosing Annotation Errors and Refining Emotion Taxonomy**: Beyond inter- and intra-class interpretability, AURA facilitates annotation error diagnosis, as shown in Fig. 3 (left), where it uncovers mislabels between "sadness" and "neutral" by exposing mismatches between facial cues and labels. Moreover, by capturing subtle variations (e.g., mild vs. intense, pure vs. compound), AURA uses archetypes to break rigid categorical boundaries and refines the seven-class taxonomy into finer, semantically coherent subsets. This yields a more meaningful organization of the affective space.

**FER Latent Space Representation**: Compared with CLIP and POSTER++ (Mao et al., 2025), which remain entangled or only partially separable, AURA produces disentangled and semantically coherent archetypes on AffectNet-8 and RAF-DB. The model adaptively discovers around 100 archetypes (93 for AffectNet-7, 98 for AffectNet-8), reflecting the need for fine-grained coverage of diverse expression categories. **VA Regression**: In AffectNet-VA, archetype allocation follows the distribution of emotional intensities, with dense coverage in moderate arousal and frequent valence regions. AURA converges to 97 archetypes for valence and 98 for arousal, ensuring comprehensive representation of the continuous affective space. **AU Detection**: In DISFA, AUs exhibit clear separations between strong and weak activations. Unlike FER and VA, AU detection focuses on localized facial details, and AURA accordingly employs fewer archetypes (41 for EmotioNet, 36 for DISFA), achieving compact yet discriminative modeling (see more detailed analysis in Appendix C).

## 4.3 ABLATION STUDY

As reported in Table 2, where CLIP with a linear head (Row 1) serves as the baseline. Removing the VAS Projector (Row 2) exposes redundant features and hurts performance, while disabling the Archetype Contextualization Module (Row 3) prevents modeling archetype relations and causes drops. VAS Embedding and archetype supervision (Row 4/5) both prove essential for convergence and separation. Without Adaptive Archetype Regularization (Row 6), archetypes collapse; removing compactness, separation, or diversity (Row 7/8/9) degrades structure, with compactness/separation most critical. Varying projection dimension (Rows 10–12) shows smaller dimensions suppress noise, whereas larger ones reintroduce irrelevant features. Finally, feature granularity (Rows 13–14) indicates FER/VA favor global features for efficiency, while AUD benefits from patch-level detail.

Table 2: Ablation study of AURA components across four datasets.

| Index | Configuration | w/o / Variant | RAF-DB | AffectNet-8 | DISFA | AffectNet-VA |
|---|---|---|---|---|---|---|
| 0 | **AURA (full model)** | – | **94.0** | **65.2** | **66.9** | **74.1** |
| 1 | CLIP Embedding + Classifier | AURA | 85.3 | 57.3 | 55.2 | 63.4 |
| 2 | Use CLIP Embedding for Archetype Discovering | VAS Projector | 88.1 | 59.3 | 58.1 | 66.2 |
| 3 | Remove Archetype Contextualization Module | AC Module | 92.7 | 63.1 | 64.2 | 73.7 |
| 4 | No Discriminative for VAS Embedding | $\mathcal{L}^{\text{VAS}}$ | 93.5 | 63.4 | 63.9 | 72.2 |
| 5 | No Discriminative for Archetype | $\mathcal{L}^{\text{Dis}}$ | 89.3 | 61.6 | 60.2 | 69.8 |
| 6 | No Adaptive Archetype Regularization | $\mathcal{L}^{\text{Reg}}$ | 92.3 | 62.1 | 64.3 | 71.2 |
| 7 | No Intra-class Compactness Regularization | $\mathcal{L}_{\text{intra}}/\mathcal{L}_{\text{repul}}$ | 90.2 | 61.2 | 61.7 | 73.1 |
| 8 | No Inter-class Separation Regularization | $\mathcal{L}_{\text{inter}}/\mathcal{L}_{\text{attr}}$ | 90.8 | 62.1 | 60.0 | 73.9 |
| 9 | No Archetype Diversity Regularization | $\mathcal{L}_{\text{div}}$ | 92.7 | 63.1 | 65.2 | 72.3 |
| 10 | | 16 | 91.3 | 60.2 | 61.5 | 69.0 |
| 11 | VAS Dim | 32 | **94.0** | **65.2** | **66.9** | **74.1** |
| 12 | | 64 | 93.1 | 64.9 | 64.1 | 73.5 |
| 13 | CLIP Feature Selection | Global-level | 94.0 | **65.2** | 62.3 | **74.1** |
| 14 | | Patch-level | **94.3** | 64.7 | **66.9** | 73.5 |

## 5 CONCLUSION

We introduced **AURA**, which adaptively discovers visual archetypes as perceptual anchors, providing a unified framework for FER, AU detection, and VA regression. These archetypes yield compact, semantically coherent, and interpretable representations that capture both category-level distinctions and intensity variations. Beyond achieving state-of-the-art performance with high efficiency, AURA also enables fine-grained interpretability, supporting error diagnosis and taxonomy refinement, thereby establishing archetypes as a powerful foundation for affective understanding.

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

## A  EXPERIMENTS

### A.1  DATASETS

**AffectNet-7/8 & AffectNet-VA** Mollahosseini et al. (2017): AffectNet is an in-the-wild database that contains around 400K images manually annotated for 6 basic expressions, as well as neutral and contempt. For our work, we utilize the manually annotated images with the 7/8 expressions category to ensure alignment with other expression datasets. AffectNet-VA provides VA annotations in the range of [-1, 1], making it suitable for dimensional affect analysis. The training set of this database consists of around 321K images and the validation of 5K. The validation set is balanced across the different expression categories. **RAF-DB (Real-world Affective Faces Database)** Li et al. (2017): RAF-DB is an in-the-wild database that contains approximately 15,000 facial images, manually annotated for 7 basic expressions. **DISFA (Denver Intensity of Spontaneous Facial Action)** Mavadati et al. (2013): DISFA is a lab controlled database consisting of videos from 27 subjects, each with approximately 5000 frames. Each frame is annotated with AU intensities on a six-point discrete scale (0–5). For consistency in AU detection tasks, we binarize the annotations, assigning a value of 1 to AU intensities greater than 2 and a value of 0 otherwise. The dataset includes annotations for 8 AUs (1, 2, 4, 6, 9, 12, 25, 26). **EmotioNet** Fabian Benitez-Quiroz et al. (2016) consists of over 45K in-the-wild facial images, where we follow the official split and use the 11 most frequent AUs for training and evaluation.

## A.2 IMPLEMENTATION DETAILS

Our AURA framework is implemented in PyTorch and trained on an NVIDIA A100 GPU. For data preprocessing, all input images are first cropped to facial regions and then resized to the CLIP-supported resolution. The CLIP visual encoder is a frozen, pre-trained model from OpenAI. Image or video frame features are extracted once using this encoder, after which all training and inference are performed purely at the feature level, eliminating the need to repeatedly invoke CLIP during optimization. We adopt the AdamW optimizer with a learning rate of $1 \times 10^{-4}$ across all datasets. To enhance generalization, a dropout rate of 0.2 is applied to both the global-level and patch-level visual projectors. For all datasets, the loss weights are set as $\lambda_{\text{Proj}} = \lambda_{\text{Arc}} = \lambda_{\text{Contx}} = 1$, ensuring balanced contributions from projection, visual archetype optimization, and refinement terms. Similarly, we set $\beta = 1$ to assign equal importance to the archetype update and commitment penalty in the vector quantization loss.

## A.3 EVALUATION PROTOCOLS

We adopt task-specific evaluation metrics to ensure fair and meaningful performance comparisons.

**Facial Expression Recognition (FER).** For FER, we report the classification accuracy (ACC), defined as:

$$\text{ACC} = \frac{1}{N} \sum_{i=1}^{N} \mathbb{I}\left(\hat{y}_i = y_i\right), \tag{19}$$

where $N$ is total number of samples, $y_i$ is the ground-truth label, $\hat{y}_i$ is predicted label, and $\mathbb{I}(\cdot)$ is the indicator function.

**Valence-Arousal (VA) Estimation.** For VA estimation, we use the Concordance Correlation Coefficient (CCC) for both valence ($v$) and arousal ($a$), defined as:

$$\text{CCC}(x, y) = \frac{2\rho_{xy}\sigma_x\sigma_y}{\sigma_x^2 + \sigma_y^2 + (\mu_x - \mu_y)^2}, \tag{20}$$

where $\rho_{xy}$ is the Pearson correlation coefficient between the predicted values $x$ and ground truth $y$, $\mu_x$ and $\mu_y$ are the means, and $\sigma_x$ and $\sigma_y$ are the standard deviations. The final CCC score is computed as the average of $\text{CCC}_v$ and $\text{CCC}_a$:

$$\text{CCC}_{\text{VA}} = \frac{\text{CCC}_v + \text{CCC}_a}{2}. \tag{21}$$

**Action Unit Detection (AUD).** For AUD, we compute the F1-score for each Action Unit (AU) independently:

$$\text{F1}_k = \frac{2 \cdot \text{Precision}_k \cdot \text{Recall}_k}{\text{Precision}_k + \text{Recall}_k}, \tag{22}$$

where $\text{Precision}_k = \frac{\text{TP}_k}{\text{TP}_k + \text{FP}_k}$ and $\text{Recall}_k = \frac{\text{TP}_k}{\text{TP}_k + \text{FN}_k}$ for AU $k$. We further report the average F1-score across all $K$ AUs:

$$\text{F1}_{\text{avg}} = \frac{1}{K} \sum_{k=1}^{K} \text{F1}_k. \tag{23}$$

## B ARCHETYPE RESET MECHANISM.

To avoid archetype collapse, we introduce a usage-aware reset mechanism that periodically reinitializes underutilized archetypes based on their global selection frequency.

*Global Usage Tracking:* Let the codebook be denoted as $\mathcal{C} = \{\mathbf{e}_1, \ldots, \mathbf{e}_N\}$, where each $\mathbf{e}_i \in \mathbb{R}^d$ is a learnable archetype. During training, we record the global usage count vector $\mathbf{u} = [u_1, \ldots, u_N] \in \mathbb{N}^N$, where $u_i$ counts the total number of times $\mathbf{e}_i$ was selected as the nearest archetype over all training steps. We define the normalized usage ratio for each code vector as: $\alpha_i = \frac{u_i}{\sum_{j=1}^{N} u_j}, \quad \forall i =$

$1, \ldots, N$, We then define a fixed threshold $\tau \in (0,1)$ (e.g., $\tau = 0.01$), and identify the set of underutilized codes: $\mathcal{P}_{\text{reset}} = \{i \mid \alpha_i < \tau\}$.

***Archetype Reset***: For each $i \in \mathcal{P}_{\text{reset}}$, we sample a new feature vector $\mathbf{f}_i \in \mathbb{R}^d$ from the current training batch and reinitialize the archetype as:

$$\mathbf{e}_i \leftarrow \mathbf{f}_i + \boldsymbol{\xi}_i, \quad \text{where } \boldsymbol{\xi}_i \sim \mathcal{N}(0, \sigma^2 \mathbf{I}),$$

with $\sigma > 0$ denoting a small Gaussian noise level used to encourage diversity. Alongside the update of $\mathbf{e}_i$, we reset all related accumulators: $u_i \leftarrow 0, \quad \mathbf{c}_i \leftarrow \mathbf{e}_i, \quad n_i \leftarrow 0$, where $\mathbf{c}_i$ denotes the accumulated cluster mean for archetype $i$ and $n_i$ is the cluster size (i.e., the count of features assigned to $\mathbf{e}_i$). Once all underutilized archetypes are updated, we reset the entire usage counter to zero: $\mathbf{u} \leftarrow \mathbf{0}$. This reset mechanism ensures that the codebook dynamically adapts to the evolving data distribution and avoids stagnation due to unused or outdated archetypes.

## C  ANALYSIS FOR LEARNED ARCHETYPES

We analyze the archetypes learned by AURA to understand their structure, distribution, and interpretability across multiple affective tasks. Our study covers (i) comparison with conventional classification model, (ii) error diagnosis and taxonomy refinement, (iii) spatial organization in arousal–valence space, (iv) allocation patterns in Action Unit spaces, and (v) quantitative cross-task statistics. The results show that AURA adaptively allocates representational capacity according to data distribution and emotional complexity, yielding both higher performance and more interpretable affective representations.

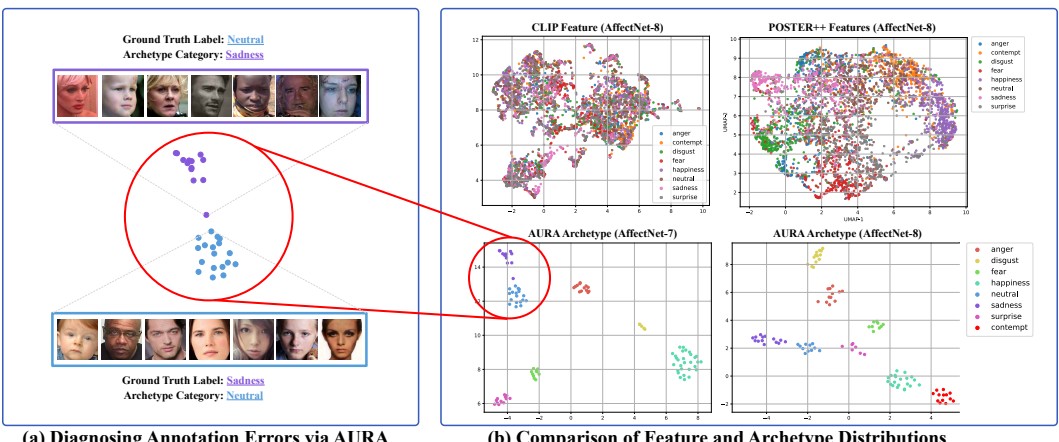

(a) Diagnosing Annotation Errors via AURA    (b) Comparison of Feature and Archetype Distributions

Figure 4: UMAP visualization of archetypes, original CLIP visual features, and POSTER++ features for the AffectNet-7/-8 facial expression recognition task. (a) Diagnosis of annotation errors using AURA; (b) Visualization of feature distributions.

### C.1  EMOTION REPRESENTATION ADVANTAGE OF AURA

*AURA vs. Conventional Classification Models:* To assess the advantages of AURA over ***conventional label-supervised classification***, which optimizes representations directly under ground-truth labels, we visualize and compare three types of learned features on the AffectNet-8 test set (Fig. 6 (b)): AURA archetypes, original CLIP features, and POSTER++ features. Our observations are as follows: **(i) Original CLIP features** are highly entangled in the affective space, yielding poor emotional separability. **(ii) POSTER++** alleviates some entanglement and improves separability, but many samples remain intertwined and the learned features still lack semantic interpretability. **(iii) AURA archetypes**, in contrast, produce highly distinct and disentangled clusters with strong semantic coherence. These results demonstrate that AURA not only surpasses conventional objectives quantitatively but also yields qualitatively more interpretable and cognitively consistent affective representations.

## C.2    Diagnosing Annotation Errors and Refining Emotion Taxonomy via AURA

We conducted an in-depth examination of the learned AURA archetypes and their associated emotion images, and found that, beyond offering inter- and intra-class interpretability (as illustrated in Fig. 3 of the main paper), AURA also serves as an effective tool for diagnosing annotation errors (as illustrated in **Fig. 6 (a)**). Upon thorough inspection, we observe that the AffectNet dataset contains a substantial number of compound expressions, which are inherently challenging to differentiate during the annotation process and therefore susceptible to mislabeling. Thanks to our semantic interpretability of AURA, we are able to systematically probe the samples assigned to each archetype, enabling precise *analysis, explanation, and error diagnosis*.

As illustrated in **Fig. 6(a)**, we identify two closely related archetypes corresponding to "sadness" and "neutral". Closer inspection of the images assigned to the "**sadness**" archetype, despite being labeled as "neutral" in the ground truth, reveals consistently sorrowful expressions characterized by knitted brows with pronounced glabellar lines, drooping eyelids, a dull gaze, and downward-turned, compressed lips. Conversely, the images mapped to the "**neutral**" archetype, though annotated as "sadness", clearly exhibit neutral facial cues, including level eyebrows, relaxed eyelids, a steady forward gaze, and lips at rest without curvature.

Notably, ***AURA refines the conventional seven-class emotion taxonomy into finer, semantically coherent subsets***, enabling more accurate grouping of visually similar expressions. Such refinement allows AURA to capture subtle variations within a single emotion class, distinguishing, for example, between mild and intense expressions or between pure and compound emotions. This finer-grained partitioning not only improves the structural organization of the affective space but also facilitates the identification of borderline or ambiguous cases that are often misclassified under rigid categorical schemes. By transcending the limitations of hard class boundaries, AURA provides a more continuous and interpretable representation of emotions, thereby enhancing both the semantic clarity of the learned features and the reliability of emotion annotations in large-scale datasets.

## C.3    Archetype Analysis in Arousal–Valence Space

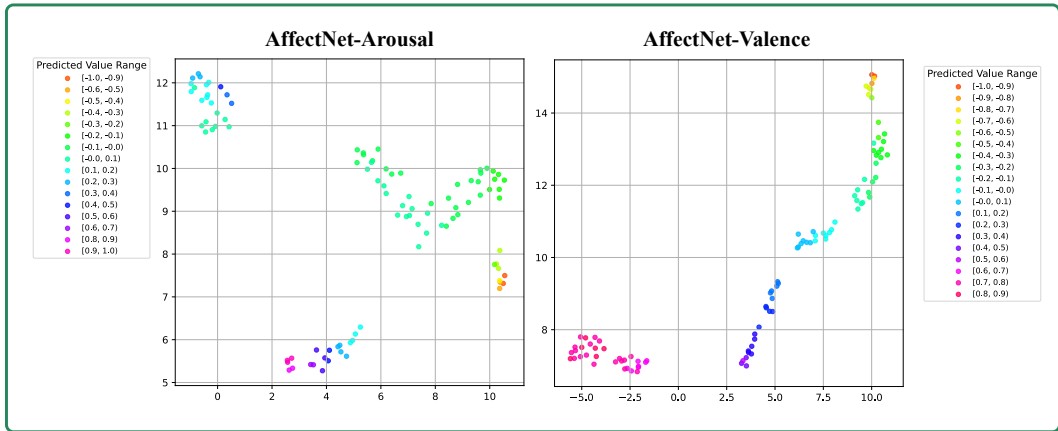

Figure 5: Valence and Arousal Prototye distribution visualization for AffectNet-VA.

**AURA For Arousal:**    A detailed examination of the AURA archetypes in the arousal dimension reveals a distinct spatial clustering pattern that aligns well with the underlying distribution of emotional intensities in the dataset. Specifically, the archetypes aggregate into four primary clusters: those corresponding to arousal values between $-1.0$ and $-0.5$ are concentrated in the lower right region (depicted by orange to yellow-green hues) comprising 9 archetypes; the range $-0.5$ to $-0.1$ forms a cluster in the mid-right region (yellow-green to cyan) containing 48 archetypes; arousal values from $-0.1$ to $0.3$ cluster in the upper left area (cyan to deep blue) with 23 archetypes; finally, values from $0.0$ to $1.0$ group near the central bottom area, comprising 20 archetypes.

This distribution reflects the natural emotional landscape captured in the dataset, where the majority of arousal values fall within the moderate range of approximately $[-0.3, 0.3]$. Emotions beyond this

Table 3: Statistics of assigned archetypes across different datasets and tasks. Each row reports the number of assigned archetypes (**Assigned Prot.**), their usage range (min–max, **Prot. Usage**), the total number of samples matched to these archetypes (**Prot. Sample Num.**), and the total number of samples in the dataset (**Sample Num.**).

| RAF-DB | | | | |
| --- | --- | --- | --- | --- |
| **Expression** | **Assigned Prot.** | **Prot. Usage** | **Prot. Sample Num.** | **Sample Num.** |
| anger | 11 | 12–566 | 710 | 705 |
| disgust | 9 | 17–288 | 751 | 717 |
| fear | 6 | 18–169 | 287 | 281 |
| happiness | 28 | 19–1788 | 4735 | 4772 |
| neutral | 19 | 8–1288 | 2417 | 2524 |
| sadness | 12 | 11–999 | 2009 | 1982 |
| surprise | 15 | 14–841 | 1362 | 1290 |
| **AffectNet-VA (Arousal / Valence)** | | | | |
| **Bin Range** | **Assigned Prot.** | **Prot. Usage** | **Prot. Sample Num.** | **Sample Num.** |
| $-1.0--0.7$ | 1 / 3 | 2341 / 1739–6783 | 2341 / 10522 | 3716 / 17989 |
| $-0.7--0.4$ | 3 / 8 | 1013–5795 / 6741–7719 | 9206 / 36362 | 12372 / 31838 |
| $-0.4--0.1$ | 28 / 13 | 1733–8666 / 1258–7383 | 63836 / 54243 | 52069 / 36753 |
| $-0.1-\ 0.1$ | 34 / 20 | 1389–7014 / 949–6734 | 71999 / 36927 | 99781 / 50891 |
| $0.1-\ 0.4$ | 22 / 17 | 1113–12569 / 154–3948 | 92903 / 20537 | 74212 / 29866 |
| $0.4-\ 0.7$ | 7 / 15 | 1178–3846 / 747–8513 | 21585 / 45156 | 30415 / 66280 |
| $0.7-\ 1.0$ | 5 / 24 | 5370–9079 / 1255–5565 | 28220 / 86343 | 21845 / 60793 |
| **DISFA (AU12 / AU25)** | | | | |
| **Bin Range** | **Assigned Prot.** | **Prot. Usage** | **Prot. Sample Num.** | **Sample Num.** |
| $0.0-\ 0.3$ | 32 / 29 | 471–5819 / 538–7391 | 62970 / 54047 | 65819 / 55054 |
| $0.3-\ 0.5$ | 1 / 2 | 1226 / 141–2440 | 1226 / 2581 | |
| $0.5-\ 0.7$ | 2 / 0 | 111–963 / 0 | 1074 / 0 | 21391 / 32156 |
| $0.7-\ 1.0$ | 9 / 13 | 411–6869 / 769–4588 | 21940 / 30582 | |

range correspond to intensely high or low arousal states, which are less frequently represented in the data and therefore require fewer archetypes for effective modeling. Conversely, the $[-0.3, 0.3]$ interval encompasses typical human emotional intensity, exhibiting rich intra-class variability that necessitates a denser population of archetypes to capture subtle distinctions. For instance, within this moderate arousal range, expressions may vary from calm attentiveness to mild agitation, each distinguished by nuanced facial cues that AURA archetypes effectively encode.

**AURA For Valence:** Turning to the valence dimension, the archetypes are distributed almost uniformly across the entire $[-1, 1]$ spectrum, with notable concentration in the intervals $[0.6, 1.0]$ and $[-0.5, 0.0]$, which are represented by 31 and 23 archetypes respectively. This allocation corresponds closely with the empirical distribution of valence in the dataset, where highly positive and mildly negative emotional states are more prevalent. The uniform spread and selective densification of archetypes indicate that AURA adapts dynamically to the data's statistical properties, providing finer granularity in emotionally significant regions while maintaining coverage across the full valence range.

Collectively, these findings underscore AURA's capacity to model the continuous valence-arousal affective space with both granularity and efficiency. By allocating archetypes in accordance with the natural distribution and complexity of emotional expressions, AURA achieves a balance between representational compactness and discriminative power, thereby enhancing interpretability and supporting nuanced emotion analysis.

### C.4 ANALYSIS OF ARCHETYPES IN AU SPACES

In this section, we visualize the learned AURA archetypes for four representative Action Units: AU4, AU12, AU25, and AU26. Across these AUs, a consistent pattern emerges whereby strong activations are associated with relatively few archetypes, while weak or absent activations correspond to a larger number of archetypes. This distribution aligns well with established domain knowledge: strongly activated AUs tend to exhibit more distinctive facial patterns, warranting compact and focused archetype representation, whereas weakly activated or inactive AUs reflect greater variability in appearance, thus requiring a more diverse set of archetypes to capture the underlying heterogeneity.

Despite this general trend, notable differences arise among the four AUs. For AU25, archetypes corresponding to strong activation levels (0.8–1.0) cluster densely in the upper-right region of the latent space, whereas weaker activations (0.1–0.4) concentrate in the lower-right region. This clear spatial segregation validates the discriminative power of AURA archetypes, as AU25's strong activation typically signifies expressions of happiness, while its weaker activation corresponds to

distinct emotional states such as disgust or contempt, underscoring AURA's capacity to capture fine-grained affective differences.

Similarly, for AU4 and AU26, strongly activated archetypes (activation levels between 0.6 and 1.0) are tightly clustered in the upper-left region, contrasting with other activation levels aggregated in the lower-right region. This spatial dichotomy reflects AURA's robust ability to sharply distinguish between active and inactive AU states.

In the case of AU12, the archetypes corresponding to moderate (0.4–0.7) and strong (0.7–1.0) activations form a contiguous cluster. This pattern is consistent with the known physiological characteristics of AU12, which often manifests with subtle gradations of activation due to the underlying facial muscle movements involved. Such nuanced clustering illustrates AURA's sensitivity to the fine-scale variations inherent in AU12 activation levels.

Overall, these findings demonstrate that AURA archetypes effectively model the complex distribution of AU activations, balancing compactness for strongly activated units with diversity for weaker activations, thereby capturing both the discriminative and variable nature of facial action units in a semantically meaningful manner.

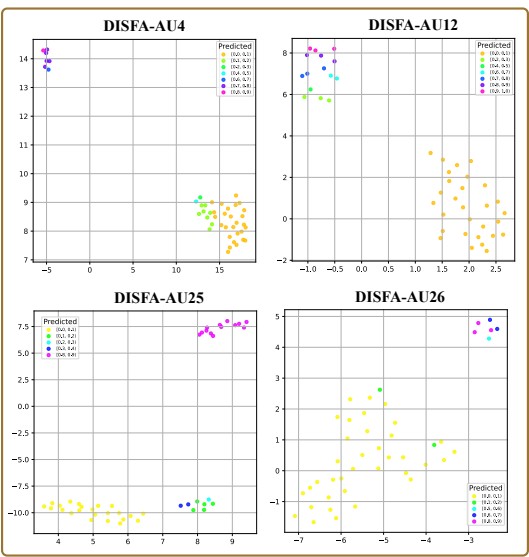

Figure 6: DISFA Archetypes distribution visualization.

## C.5 Quantitative Analysis of Archetype Distribution Across Tasks

We present a quantitative analysis of archetype allocation patterns across three representative affective tasks: categorical facial expression recognition (RAF-DB), continuous arousal–valence estimation (AffectNet-VA), and action unit detection (DISFA). The statistics in Table 3 summarize the number of assigned archetypes (**Assigned Prot.**), their usage ranges (**Prot. Usage**), the total number of samples matched to these archetypes (**Prot. Sample Num.**), and the total dataset sample counts (**Sample Num.**). This quantitative view allows us to interpret how the AURA mechanism distributes representational capacity across different affective states, intensities, and data densities.

**RAF-DB (Expression Recognition):** Archetype allocation varies substantially across the seven expression categories. High-frequency and visually diverse categories such as *happiness* (28 archetypes, usage range: 19–1788) and *neutral* (19 archetypes, 8–1288) receive a larger number of archetypes with broad usage spans, indicating high intra-class variability. Conversely, categories such as *fear* (6 archetypes, 18–169) and *disgust* (9 archetypes, 17–288) have fewer archetypes and narrower ranges, reflecting lower diversity and sample counts. *Sadness* and *surprise* fall in between, with moderate archetype counts but concentrated usage, suggesting more homogeneous visual patterns.

**AffectNet-VA (Arousal / Valence Estimation):** In the continuous affective space, archetype allocation strongly correlates with data density. Extreme affective regions (e.g., $-1.0$–$-0.7$, $0.7$–$1.0$)

exhibit fewer archetypes (1–5 for arousal, 3–24 for valence) and lower matched sample counts, due to the scarcity of highly polarized emotions in the dataset. In contrast, the central regions (e.g., −0.1–0.1, 0.1–0.4) receive the largest number of archetypes (up to 34 for arousal, 20 for valence) and significantly higher sample counts, capturing subtle variations in near-neutral affective states. This aligns with AffectNet's known bias toward mild or mixed emotions.

**DISFA (Action Unit Detection):** For AU-based modeling, archetype allocation distinguishes between inactive/low-intensity and highly active facial muscle states. In AU12, the 0.0–0.3 range dominates with 32 archetypes and 62,970 matched samples, while the mid-intensity range (0.3–0.5) is covered by only a single archetype, indicating rare occurrences. High-intensity activations (0.7–1.0) have fewer archetypes (9 for AU12, 13 for AU25) but disproportionately high sample counts, suggesting these expressions, while less visually diverse, are relatively frequent in the dataset. Notably, AU25 has no archetypes in the 0.5–0.7 range, implying low occurrence or ambiguity in this activation intensity.

This quantitative view highlights that archetype allocation in AURA is inherently data-adaptive. Tasks and affective states with high visual diversity or dense sample distributions receive more archetypes with wider usage ranges, while homogeneous or rare states are represented by fewer archetypes with concentrated usage. This property ensures both representation efficiency and strong discriminative capacity across heterogeneous affective modeling scenarios.

# D  FAILURE CASE ANALYSIS

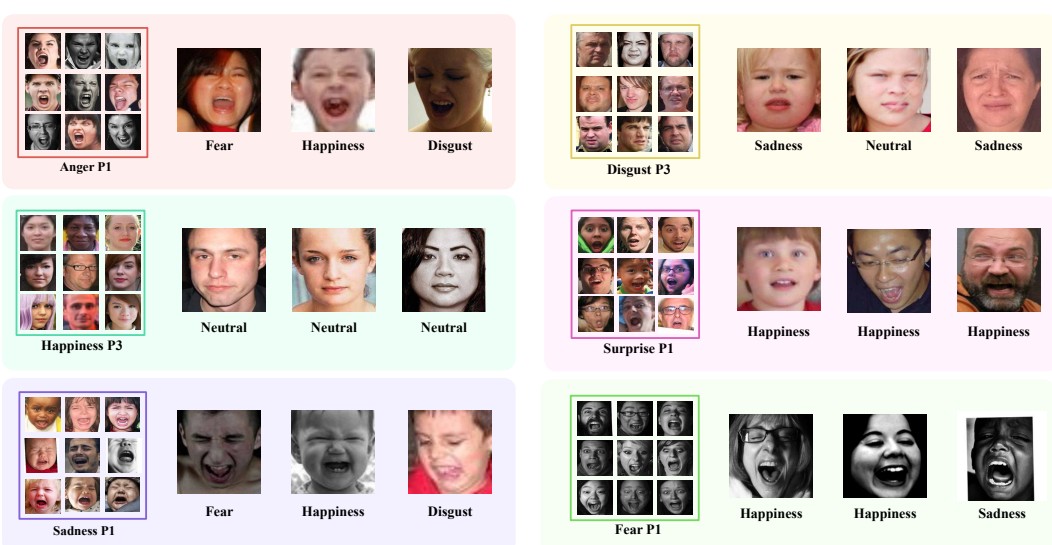

Figure 7: Failure Case Analysis. For each panel, the **Left** column shows correctly labeled samples are aligned with the corresponding expression archetype, while the **Right** column shows samples assigned to this archetype despite belonging to other expression classes.

To further evaluate the behavioral characteristics of AURA, we conduct a dedicated failure-case analysis, with results shown in Figure 7. For each panel, the **Left** block presents the support images that are aligned with a particular expression archetype, while the **Right** side displays misclassified samples that were assigned to this archetype despite belonging to different ground-truth classes. This setup enables simultaneous examination of **(i) how the model behaves on individual samples and (ii) what semantic structures lead to these error**s. By comparing each mispredicted sample with the defining archetype, we can clearly identify which facial configuration the model relied on (e.g., mouth shape, eye tension, or brow contraction), thereby revealing the semantic basis of the model's mistakes.

Across all panels, a clear and consistent pattern emerges: although the categorical prediction is wrong, the assigned archetype remains semantically justified because the misclassified faces share the **same fine-grained facial-muscle configuration** as the archetype's support set. Crucially, these archetypes

do not simply correspond to **"open-mouth"** expressions in general; each archetype encodes a distinct sub-pattern of facial activation. For example, **Anger P1** captures an expression characterized by a maximally forceful, ***lower-face*** (dominant contraction, a wide, tense mouth opening with strong jaw engagement), so samples with an equally forceful lower-face stretch are naturally drawn to this archetype, even if their categorical label is Fear or Disgust. In contrast, **Sadness P1** reflects a different structural signature: although the mouth is open, the expression is dominated by ***upper-face*** tension (furrowed brows, nasal contraction, and a drooping eye region). Misassigned samples in this panel exhibit precisely this upper-face configuration, explaining why the model anchors them to this archetype. **Fear P1**, on the other hand, is defined by a high-intensity, mouth-stretched configuration ***without pronounced brow contraction***, often manifested in ***grayscale*** images within the dataset; misclassified Happiness or Sadness samples assigned to Fear P1 share this same "wide-mouth, minimal-brow-movement" structure. These observations show that AURA's archetypes capture fine-grained, physiologically meaningful facial patterns, and that mispredictions occur not because the model is confused arbitrarily, but because the sample's *local facial configuration* aligns more closely with a specific archetypal mode than with its discrete ground-truth label.

In the **Disgust P3** panel, the archetype is characterized by nose wrinkling, ***raised upper lip***, and narrowed eyes. Samples labeled as Sadness or Neutral are assigned to this archetype because they exhibit a very similar nose–mouth configuration, suggesting label ambiguity or overlapping expression cues in the dataset. In **Happiness P3**, the archetype reflects subtle, ***low-intensity smiles*** close to Neutral faces. The misclassified Neutral samples assigned to this archetype have very ***mild lip corners*** raised and relaxed upper faces, revealing that the boundary between low-intensity Happiness and Neutral is intrinsically fuzzy. Finally, **Surprise P1** collects wide-eyed, mouth-open faces; the mispredicted Happiness examples assigned to it also display strong ***"wow"-like*** configurations, again showing that the model is grouping samples by a coherent facial pattern rather than arbitrary noise.

Across all panels, AURA's failure cases reveal a consistent and meaningful pattern: even when the discrete class prediction is incorrect, the assigned archetype remains semantically well-aligned with the facial structure of the input. This is because the misclassified faces share highly similar AU configurations, intensity patterns, and local geometry with the archetype's support set.

The misassigned samples naturally fall into these archetypes because their fine-grained facial structure more closely matches the archetype's learned configuration than their categorical label suggests. Importantly, such mismatches often arise from dataset statistics: if a structural pattern appears predominantly in one class (e.g., scream-like faces in Anger) and is underrepresented in others (e.g., scream-like Happiness), the model will gravitate toward the archetype that best captures that structure. Rather than being a weakness, this demonstrates the strength of our weakly supervised archetype formulation: **AURA prioritizes genuine facial-structure similarity over noisy or ambiguous labels, offering per-sample insight into which semantic mode the model activated and why.**

# E  SENSITIVITY TO THE CHOICE OF $K_{\max}$

This section clarifies the distinction between the predefined upper bound of archetypes and the effective number that AURA ultimately employs. AURA differentiates between two quantities: the predefined upper bound $K_{\max}$, typically set between 150 and 400, and the stable number of active archetypes $K_{\text{stable}}$ that emerges automatically during training. The value of $K_{\text{stable}}$ is not determined by $K_{\max}$; instead, $K_{\max}$ is chosen to be over-complete to ensure sufficient representational capacity, while the model identifies a much smaller and semantically meaningful subset of archetypes according to the task. Empirically, AURA converges to approximately 100 active archetypes for expression recognition on AffectNet and RAF-DB, and around 40 for AU recognition on EmotioNet, regardless of the initial choice of $K_{\max}$.

The adaptivity of AURA arises from two key components: the Adaptive Archetype Regularization and the Archetype Contextualization Module. Together, these mechanisms encourage informative archetypes to receive substantial assignments while suppressing redundant ones. Let the archetype codebook be $\mathcal{C} = \{e_1, \ldots, e_{K_{\max}}\}$. During training, AURA maintains a global usage counter $u_k$ for each archetype, recording how often $e_k$ is selected as the primal archetype. A normalized usage ratio is computed as $\alpha_k = u_k / \sum_{j=1}^{K_{\max}} u_j$. Archetypes with usage ratio below a threshold $\tau$ (e.g., $\tau = 0.01$) are considered under-utilized. During the early stage of optimization (first 20–30 epochs), an Archetype Reset Mechanism reinitializes the embeddings of under-utilized archetypes to avoid

premature collapse of dictionary capacity. After this warm-up stage, the usage stabilize, producing a consistent $K_{\text{stable}}$ that remains largely invariant to $K_{\text{max}}$. At inference time, archetypes with $\alpha_k < \tau$ are pruned, yielding a compact and interpretable dictionary.

To examine robustness with respect to $K_{\text{max}}$, we conduct a sensitivity analysis on RAF-DB with $K_{\text{max}} \in \{150, 200, 300, 400\}$. As summarised in Table 4, the resulting number of active archetypes remains within a narrow range (99-102) for all settings. Model accuracy varies within only $0.2\%$, convergence epochs remain comparable, and the class-wise archetype distribution exhibits high consistency. The computational cost increases moderately for larger $K_{\text{max}}$ due to the expanded over-complete dictionary, but this does not affect the effective number of utilized archetypes. These results demonstrate that AURA automatically identifies a suitable number of archetypes and is robust and insensitive to the initial value of $K_{\text{max}}$ in terms of performance, convergence behaviour, and the granularity–efficiency trade-off.

Table 4: Sensitivity of AURA to the choice of $K_{\text{max}}$ on RAF-DB.

| $K_{\text{max}}$ | $K_{\text{stable}}$ | ACC (%) | Convergence Epoch | Archetype Distribution | GFLOPs |
|---|---|---|---|---|---|
| 150 | 100 | 94.1 | 153 | Anger 11 / Disgust 9 / Fear 6 / Happiness 28 / Neutral 19 / Sadness 12 / Surprise 15 | 0.26 |
| 200 | 99 | 94.0 | 147 | Anger 11 / Disgust 9 / Fear 6 / Happiness 26 / Neutral 20 / Sadness 12 / Surprise 15 | 0.29 |
| 300 | 102 | 94.2 | 158 | Anger 11 / Disgust 10 / Fear 6 / Happiness 29 / Neutral 18 / Sadness 13 / Surprise 15 | 0.44 |
| 400 | 101 | 94.1 | 155 | Anger 12 / Disgust 10 / Fear 6 / Happiness 29 / Neutral 19 / Sadness 13 / Surprise 12 | 0.48 |

## F  ENCODER-AGNOSTIC BEHAVIOR OF AURA

To isolate the contribution of the proposed archetype mechanism from the representational priors of CLIP, we extend our study by conducting a dedicated evaluation using both **CLIP** (Radford et al., 2021) and **DINO** (Caron et al., 2021). In particular, DINO serves as a purely self-supervised vision backbone without any text–image alignment, providing a controlled testbed to determine whether AURA's improvements originate from CLIP's semantically aligned visual space or from the archetype modeling itself. This analysis allows us to rigorously disentangle the effects of encoder pretraining and the structural advantages introduced by AURA.

To ensure a fair and comprehensive comparison, we evaluate each encoder under two parallel configurations. For DINO, we first consider **DINO FT (full fine-tuning)**, where the entire DINO student network is optimized jointly with task-specific heads following official training protocols. All backbone parameters are trainable in this setting. We then construct a second configuration, **AURA-DINO**, in which the CLIP backbone in our main AURA model is replaced by the pretrained DINO encoder, but the encoder is kept *entirely frozen*. DINO is used strictly as a feature extractor, and only AURA's archetype modules are updated during training. This frozen setting cleanly isolates the effect of archetype modeling by removing any benefits from encoder adaptation.

For completeness, we perform the same two configurations using CLIP. In the **CLIP FT** condition, the full CLIP visual encoder is fine-tuned end-to-end along with the task heads. In the **AURA-CLIP** setting, the official pretrained CLIP encoder is kept *frozen*, and AURA operates solely on top of its fixed visual embeddings. By aligning CLIP and DINO under identical experimental protocols, this cross-encoder evaluation enables a controlled investigation of whether AURA's gains are tied to CLIP's vision–language alignment or generalize across fundamentally different pretraining paradigms.

The results, summarized in Table **??**, show that AURA delivers *strong and consistent performance gains* across both CLIP and DINO. Notably, despite relying on *frozen* DINO features without any fine-tuning of backbone parameters, AURA-DINO achieves substantial improvements over the fully fine-tuned DINO baseline: +4.3% accuracy on RAF-DB, +6.6 CCC on AffectNet-VA, and +4.7 F1 on EmotioNet. While absolute performance under DINO is naturally lower than under CLIP—reflecting CLIP's multimodal supervision and richer semantic priors—the *relative* gains contributed by AURA remain highly consistent across FER, VA, and AU tasks.

An additional observation further strengthens this conclusion: although CLIP outperforms DINO in absolute terms, owing to its stronger semantic alignment, *the performance gains introduced by AURA are even larger on CLIP than on DINO*. This pattern reveals two important insights. First, the

benefits of AURA do not arise from CLIP's multimodal alignment alone, as similar gains appear with a purely visual self-supervised encoder. Second, AURA is capable of fully exploiting the representational capacity of stronger pretrained models—particularly CLIP's semantically aligned embedding space—yielding more structured latent geometries, reduced intra-class ambiguity, and improved predictive accuracy.

Overall, these findings demonstrate that AURA is fundamentally **encoder-agnostic**. Its improvements stem from the archetype-based feature structuring itself rather than any encoder-specific advantage. AURA consistently enhances a wide range of pretrained vision models by discovering canonical semantic anchors and decomposing fine-grained intra-class variability, independently of whether the underlying backbone is multimodally aligned (CLIP) or purely visual (DINO).

Table 5: Evaluating AURA with CLIP and DINO encoders. "Official FT" denotes full fine-tuning; "AURA (ours)" denotes frozen encoder + AURA.

| Method | Encoder | RAF-DB Acc | AffectNet-VA CCC | EmotioNet AU-F1 |
|---|---|---|---|---|
| CLIP FT | CLIP (finetuned) | 89.1 | 66.4 | 62.3 |
| **AURA-CLIP** | **CLIP (official frozen)** | **94.0 (+4.9)** | **74.1 (+7.7)** | **67.3 (+5.0)** |
| DINO FT | DINO (finetuned) | 88.3 | 65.4 | 60.7 |
| **AURA-DINO** | **DINO (official frozen)** | **92.6 (+4.3)** | **72.0 (+6.6)** | **65.4 (+4.7)** |

## G  SENSITIVITY ANALYSIS OF TRAINING HYPERPARAMETERS

This section provides a comprehensive and detailed analysis of the sensitivity of AURA to its major training hyperparameters. Although AURA contains several components in its loss formulation, the overall training process is intentionally designed to remain simple, stable, and fully reproducible across datasets and tasks.

### G.1  SENSITIVITY TO THE LOSS-WEIGHT COEFFICIENTS $\lambda$

This appendix provides a detailed analysis of the sensitivity of AURA with respect to the loss-weight coefficients used in the total objective

$$\mathcal{L} = \lambda_{\text{Proj}}\mathcal{L}^{\text{VAS}} + \lambda_{\text{Arc}}\mathcal{L}^{\text{Arc}} + \lambda_{\text{Contx}}\mathcal{L}^{\text{Contx}}.$$

Across all experiments, we adopt the uniform setting $\lambda_{\text{Proj}} = \lambda_{\text{Arc}} = \lambda_{\text{Contx}} = 1$, without any tuning. This simple configuration consistently produces strong results on RAF-DB, EmotioNet, and AffectNet for FER, AU detection, and VA regression, indicating that AURA does not rely on delicate loss balancing.

The reason equal weighting works is rooted in the architectural decomposition of AURA: the three loss terms act on **disjoint parameter subsets**, preventing gradient competition. The projection-supervision loss $\mathcal{L}^{\text{VAS}}$ governs only the projection head and aligns archetype mixtures with task semantics. The archetype-regularization loss $\mathcal{L}^{\text{Arc}}$ acts exclusively on the archetype dictionary, shaping geometry, sparsity, and separation. The contextual-interaction loss $\mathcal{L}^{\text{Contx}}$ applies only to the attention module that mediates cross-archetype message passing. Since each component optimizes an independent representational layer, the gradients naturally remain compatible in scale even without explicit balancing.

Within $\mathcal{L}^{\text{Arc}}$, the components $\mathcal{L}^{\text{Assign}}$, $\mathcal{L}^{\text{Dis}}$, and $\mathcal{L}^{\text{Reg}}$ operate on bounded similarity measures (cosine similarity or simplex-normalized assignments), which keeps their magnitudes comparable. These forces **act in complementary directions**—assignment encourages confident usage of archetypes, the distance term promotes geometric separation, and the regularization term stabilizes class- or score-conditioned structure. Maintaining equal weights ensures a stable and unbiased equilibrium: no component overwhelms the archetype geometry, and the system avoids collapse or overspreading. Although the form of $\mathcal{L}^{\text{Reg}}$ differs between FER, AU, and VA tasks, all variants enforce inter-

class distinction, intra-class compactness, and diversity; each is designed to operate within similar numerical ranges, further supporting robust behavior under $\lambda = 1$.

To empirically verify sensitivity, we vary each coefficient within $\lambda \in \{0.5, 1.0, 1.5\}$ and evaluate all combinations. Across all datasets, the resulting performance remains remarkably stable, with only minor fluctuations attributable to convergence speed rather than model quality. The comprehensive results are summarized in Table 6. The negligible variation confirms that AURA's training dynamics are inherently well-balanced and do not require hyperparameter tuning for loss weights.

Table 6: Comprehensive sensitivity analysis of the loss-weight coefficients $\lambda$ across FER (RAF-DB), AU (EmotioNet), and VA (AffectNet-VA).

| $(\lambda_{\text{Proj}}, \lambda_{\text{Arc}}, \lambda_{\text{Contx}})$ | RAF-DB ACC | Epoch | AU-F1 | Epoch | VA-CCC | Epoch |
|---|---|---|---|---|---|---|
| (1.0, 1.0, 1.0) **(Default)** | 94.0 | 152 | 67.3 | 136 | 74.1 | 188 |
| (0.5, 1.0, 1.0) | 93.8 | 187 | 67.1 | 165 | 74.1 | 203 |
| (1.5, 1.0, 1.0) | 94.0 | 149 | 66.8 | 138 | 74.2 | 184 |
| (1.0, 0.5, 1.0) | 94.2 | 176 | 67.1 | 148 | 73.9 | 189 |
| (1.0, 1.5, 1.0) | 94.3 | 142 | 67.2 | 136 | 74.0 | 196 |
| (1.0, 1.0, 0.5) | 93.8 | 180 | 66.9 | 164 | 74.0 | 207 |
| (1.0, 1.0, 1.5) | 94.1 | 146 | 67.2 | 122 | 74.2 | 175 |

### G.2 SENSITIVITY TO THE MARGIN PARAMETER $m$

This appendix provides a detailed analysis of the sensitivity of AURA to the choice of the margin $m$ used in the archetype regularization terms. Conceptually, the margin $m$ plays a unified role across tasks: it specifies the threshold that separates pairs that *should be close* from pairs that *should be separated*. As long as $m$ is chosen within a semantically meaningful range that is consistent with the underlying similarity or distance scale, the model remains stable and does not require fine-grained tuning.

**In the classification setting**, the inter-class separation loss penalizes pairs of class centers whose cosine similarity exceeds the margin $m$. Formally, the loss takes the form

$$\mathcal{L}_{\text{inter}} = \frac{1}{|\mathcal{S}|} \sum_{(c,c') \in \mathcal{S}} \max\left(0, \cos(\tilde{\mu}_c, \tilde{\mu}_{c'}) - m\right),$$

where $\mathcal{S}$ indexes pairs of distinct classes and $\tilde{\mu}_c$ denotes the normalized center of class $c$. In this formulation, $m$ directly defines the maximum allowable similarity between different classes: pairs with cosine similarity below $m$ incur no penalty, whereas pairs with similarity above $m$ are pushed apart. Since cosine similarities on normalized centers lie in $[0, 1]$, choosing $m$ in the range $[0.2, 0.4]$ yields a natural trade-off between enforcing sufficient separation and avoiding overly aggressive repulsion. Empirically, this interval provides a stable operating region for both FER and AU classification.

**In the regression setting**, the margin appears in the score-aware attraction and repulsion losses, which jointly control how archetypes with similar or dissimilar predicted scores are arranged in the feature space. Let $\Delta^{ij}$ denote the score difference between two archetypes and $d^{ij}$ their cosine distance. The attraction loss encourages archetypes with similar predicted scores to be close, using a hinge on $(d^{ij} - m)$, while the repulsion loss enforces separation for archetypes with divergent scores, using a hinge on $(m - d^{ij})$. In this case, $m$ plays a dual role: it defines the boundary between "similar-score pairs" and "dissimilar-score pairs", and it sets the tolerance radius within which attraction is active or beyond which repulsion becomes necessary. Because valence–arousal targets span a broader semantic range (after normalization to $[-1, 1]$), meaningful score gaps tend to be larger, and slightly smaller margins in the interval $m \in [0.1, 0.3]$ yield stable and effective behavior.

To empirically validate the above analysis, we conduct a controlled sweep over a range of margin values and evaluate performance on all three tasks. The results are reported in Table 7. As expected, performance degrades only when $m$ is set too small, which leads to excessively strong separation forces and can fragment the representation, or when $m$ is set too large, which weakens the separation

and reduces discriminability. Within the intermediate ranges described above, the accuracy, F1, and CCC metrics remain stable, confirming that AURA is insensitive to moderate changes in $m$ and does not rely on careful tuning of this hyperparameter.

Table 7: Sensitivity analysis of the margin $m$ across FER (RAF-DB), AU detection (EmotioNet), and VA regression (AffectNet-VA).

| $m$ | RAF-DB (ACC) | EmotioNet (AU-F1) | AffectNet-VA (CCC) |
|-----|--------------|-------------------|--------------------|
| 0.1 | 93.2 | 66.7 | 73.9 |
| 0.2 | 93.9 | 67.2 | **74.1** |
| 0.3 | **94.1** | **67.3** | 74.0 |
| 0.4 | 94.0 | 67.3 | 73.8 |
| 0.5 | 93.6 | 66.9 | 73.4 |

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
