# OpenReview forum: "AURA: Visually Interpretable Affective Understanding via Robust Archetypes"
_ICLR.cc/2026/Conference — Submitted to ICLR 2026_

### Official Review · Reviewer_EDGc · 2025-10-29

**Soundness:** 2
**Presentation:** 2
**Contribution:** 2
**Rating:** 4
**Confidence:** 2

**Summary:**

This paper proposes AURA, a prompt-free framework for affective understanding that operates via self-organized visual archetypes in a frozen CLIP visual space. It aims to unify classification (FER), detection (AU), and regression (VA) tasks, claiming SOTA performance, improved interpretability, and high efficiency across six benchmarks.

**Strengths:**

1. Originality: Introduces a unified visual-archetype paradigm, moving away from brittle text prompts.

2. Significance: A well-motivated approach with potential to influence affective computing and interpretable ML.

3. Experiments: Comprehensive evaluation across three tasks; ablation studies support design choices.

**Weaknesses:**

1.  **Misleading Claim of "Direct" Operation in CLIP Space:** The paper repeatedly states it operates "directly in a frozen CLIP visual space." However, the core of the method relies on a **trainable Visual Archetype-Space Projector (VAS)**, `ℱ(·)`, which transforms the original CLIP features into a new, task-specific "archetype space." This is not a direct operation but a learned adaptation. This phrasing is conceptually misleading and should be clarified to avoid confusion.

2.  **Insufficient Detail on Adaptive Archetype Mechanism:** While the "self-organized archetype discovery" is a key contribution, the process for adaptively determining the final number of archetypes per state is underspecified. The text mentions it "adaptively discovers around 100 archetypes" but provides no concrete criteria, convergence conditions, or initialization strategy for this process. The lack of reproducibility details here is a significant methodological weakness.

3.  **Inadequate Explanation for Feature Granularity Results:** The ablation study (Table 2) shows that patch-level features, intended for local AU detection, slightly outperform global features on RAF-DB (a global FER task). This counter-intuitive result is not discussed or analyzed. The authors should investigate whether this indicates that local features are unexpectedly beneficial for certain expression categories or if it reveals a more complex relationship between feature granularity and task performance that is not yet understood.

**Questions:**

1.  Please clarify the methodological description: The framework learns a projection from the CLIP space to an archetype space. Revising the "operates directly" claim would improve conceptual accuracy.
2.  What are the specific, concrete criteria or algorithm for determining the final number of archetypes `K` in the self-organizing process? How is it initialized and when does the adaptation converge?
3.  How do you explain the superior performance of patch-level features on the global FER task (RAF-DB) in your ablation study? Please provide a hypothesis or further analysis.
4.  Please provide all supplementary material (Appendices A, C) for a complete assessment of implementation details and reproducibility.

---

> ### Author Response · Authors · 2025-11-24
> **Author's Response - Thank you for your feedback (part 1)**
>
> We sincerely thank the reviewer for the thoughtful and valuable feedback. We appreciate the opportunity to improve the clarity and quality of our work. All modifications made in the manuscript and Appendix are highlighted in **Pink** and **Red**. Our detailed responses to each comment are provided below.
>
> -----------
>
> ### Weakness 1
>
> -----------
>
> > Weakness 1. **Misleading Claim of "Direct" Operation in CLIP Space:** The paper repeatedly states it operates "directly in a frozen CLIP visual space." However, the core of the method relies on a **trainable Visual Archetype-Space Projector (VAS)**, `ℱ(·)`, which transforms the original CLIP features into a new, task-specific "archetype space." This is not a direct operation but a learned adaptation. This phrasing is conceptually misleading and should be clarified to avoid confusion.
> > Question 1. Please clarify the methodological description: The framework learns a projection from the CLIP space to an archetype space. Revising the "operates directly" claim would improve conceptual accuracy.
>
> We have revised all similar expressions in the main paper and highlighted the updated text in **Pink**. The corresponding changes can be found at **Lines 016–018, 058–059, and 075–077**. We have changed similar phrasing to the more precise "projecting frozen CLIP visual embeddings into a learnable archetype space."

---

> ### Author Response · Authors · 2025-11-24
> **Author's Response - Thank you for your feedback (part 2)**
>
> ### Weakness 2
> -----------
>
> > Weakness 2: **Insufficient Detail on Adaptive Archetype Mechanism:** While the "self-organized archetype discovery" is a key contribution, the process for adaptively determining the final number of archetypes per state is underspecified. The text mentions it "adaptively discovers around 100 archetypes" but provides no concrete criteria, convergence conditions, or initialization strategy for this process. The lack of reproducibility details here is a significant methodological weakness.
> > Question 2: 1. What are the specific, concrete criteria or algorithm for determining the final number of archetypes $K$ in the self-organizing process? How is it initialized and when does the adaptation converge?
>
> To improve clarity, we have added a comprehensive analysis in main manuscripts **Section 3** and **Appendix E**, highlighted in **Red**. Our clarification is summarized in the following:
>
> **Details of the Adaptive Archetype Discovery Mechanism**
>
> - **Archetype Quantities**: In terms of the need for more explicit methodological details regarding how AURA adaptively determines the effective number of archetypes, we clarify that AURA operates with two distinct quantities: a **predefined upper bound** $K\_{\max}$ that specifies the size of an over-complete initial codebook, and a **learned stable number** of active archetypes $K\_{\text{stable}}$ that emerges naturally during training. The choice of $K\_{\max}$ (e.g., 150–400) does *not* dictate the final number of archetypes; instead, it serves only as an initialization to ensure sufficient expressive capacity. Archetypes are initialized with $\ell\_2$-normalized Gaussian vectors and uniform prior assignment probability, providing a reproducible and unbiased starting point.
> - **Archetype Usage Criterion and Reset Mechanism.** AURA determines active archetypes using a **concrete usage criterion (Appendix B)**. Let the archetypes codebook be $\mathcal{C}={e\_1,\dots,e\_{K\_{\max}}}$. During training, we track the global usage count $u\_k$ for each archetype—i.e., how often $e\_k$ is selected as the primal archetype. We compute its **normalized usage ratio** $\alpha\_k = \frac{u\_k}{\sum\_{j=1}^{K\_{\max}} u\_j}$. Archetypes with $\alpha\_k < \tau$ (e.g., $\tau = 0.01$) are considered under-utilized. In the early stage (first 20–30 epochs), any under-utilized archetype triggers the **Archetype Reset Mechanism (Appendix B)**, where its embedding $e\_k$ is reinitialized to prevent premature collapse. After this phase, the usage ratios stabilize, yielding a consistent $K\_{\text{stable}}$ (≈100 for FER, ≈40 for AU) regardless of the initial $K\_{\max}$.
> - **Convergence Conditions**: The adaptive archetype discovery process exhibits clear and easily verifiable convergence behavior. The most direct indicator is that the number of active archetypes $K\_{\text{stable}}^{(t)}$ and distribution across all label range becomes stable after the early training stage, fluctuating only minimally across subsequent epochs. Beyond this structural convergence, the training loss curve behaves as expected for a well-optimized model (gradually decreasing and reaching a steady plateau), while validation performance remains stable without oscillation. At inference time, archetypes with $\alpha\_k < \tau$ are pruned, resulting in a compact and semantically meaningful archetype dictionary.
> - **Sensitivity Experiments**: To prove AURA robustness, we include a sensitivity study (**Table 1 below**) varying $K\_{\max}$ from 150 to 400. Across this wide range, AURA consistently learns nearly identical $K\_{\text{stable}}$ (99-102 on RAF-DB), achieves accuracy within only ±0.2%, shows highly similar emotion-wise archetype distributions, and converges in comparable epochs. These results demonstrate that AURA’s adaptive mechanism is **stable, reproducible, and insensitive** to the initial over-complete dictionary size, fully addressing the reviewer’s concern regarding criteria, convergence behavior, and initialization strategy.
>
> **Table 1. Sensitivity of AURA to the choice of ($K\_{\max}$) on RAF-DB.**
>
> | $K\_{\max}$ | Stable Archetypes ($K\_\text{stable}$) | ACC (%) | Convergence Epoch | Archetypes Distribution                                                      | Computational Efficiency （GFLOPs） |
> | ---------- | ------ | ------- | ------ | ---------------- | ----------- |
> | 150  | 100         | 94.1    | 153     | Anger 11 / Disgust 9 / Fear 6 / Happiness 28 / Neutral 19 / Sadness 12 / Surprise 15   | 0.26                              |
> | 200  | 99    | 94.0    | 147    | Anger 11 / Disgust 9  / Fear 6 / Happiness 26 / Neutral 20 / Sadness 12 / Surprise 15    | 0.29        |
> | 300 | 102    | 94.2    | 158     | Anger 11 / Disgust 10 / Fear 6  / Happiness 29 / Neutral 18 / Sadness 13 / Surprise 15   | 0.44       |
> | 400 | 101  | 94.1    | 155     | Anger 12 / Disgust 10 / Fear 6  / Happiness 29 / Neutral 19 / Sadness 13 / Surprise 12   | 0.48         |

---

> ### Author Response · Authors · 2025-11-24
> **Author's Response - Thank you for your feedback (part 3)**
>
> ### Weakness 3
> -----------
>
> > Weakness 3: **Inadequate Explanation for Feature Granularity Results:** The ablation study (Table 2) shows that patch-level features, intended for local AU detection, slightly outperform global features on RAF-DB (a global FER task). This counter-intuitive result is not discussed or analyzed. The authors should investigate whether this indicates that local features are unexpectedly beneficial for certain expression categories or if it reveals a more complex relationship between feature granularity and task performance that is not yet understood.
> > Question 3. How do you explain the superior performance of patch-level features on the global FER task (RAF-DB) in your ablation study? Please provide a hypothesis or further analysis.
>
>
> **Explanation of Patch-Level vs. Global-Level Feature Performance on RAF-DB**
>
> We investigated the differences and found that the phenomenon is consistent with the **dataset-specific characteristics of facial signals**, rather than an inconsistency in the AURA framework. While patch-level features slightly outperform global features on RAF-DB. This behavior is expected for the following reasons:
> - **(1) RAF-DB contains clearer and more distinguishable local muscular cues.**
> 	RAF-DB dataset's facial regions are typically **more centrally framed, less occluded, and better aligned** compared with large-scale datasets AffectNet. Many RAF-DB expression categories (e.g., disgust, sadness, surprise) are characterized by **distinct local Action Unit activations** (e.g., nose wrinkling, brow contraction, eyelid widening). These localized patterns are precisely the type of cues that patch-level features capture effectively, enabling AURA to form slightly more discriminative archetypes. This interpretation is also consistent with the empirical results: RAF-DB achieves **over 94% accuracy**, and the performance gap between patch-level and global features is **very small**, indicating that both feature granularities already provide strong information for this relatively cleaner dataset.
> - **(2) AffectNet and AffectNet-VA exhibit higher variation, occlusion, and noise, which reduces the reliability of patch-level signals.**
> 	AffectNet includes significantly more **extreme head poses**, **partial occlusions**, **illumination inconsistencies**, and **annotation noise**. Under such conditions, individual patches often become unreliable—some facial regions may be invisible, blurred, or distorted. In contrast, global features aggregate holistic information and are naturally more robust to missing or corrupted local regions. This explains why the global representation outperforms patch-level features on AffectNet, and especially on AffectNet-VA, where continuous valence/arousal regression is inherently **global-affective** rather than driven by local AUs.
> Overall, these observations reveal **no contradiction**; instead, they demonstrate that AURA effectively **adapts to whichever feature granularity (local or global) offers stronger semantic cues** for each dataset and task. From the perspective of **utility–efficiency trade-off**, we choose the global representation as the default input for RAF-DB in our main experiments, since it offers similar accuracy with lower computational cost. Nevertheless, AURA remains fully **feature-agnostic**, and users can flexibly choose global, patch-level, or hybrid representations depending on their application needs.
>
> ---
> ### Weakness 4
>
> -----------
>
> > Weakness 4. Please provide all supplementary material (Appendices A, C) for a complete assessment of implementation details and reproducibility.
>
> In the original submission, Appendices A and C were included in the **Supplementary Material**, which may have caused confusion during review. We apologize for the inconvenience. In the revised version, we have moved all appendix content **into the main manuscript**, and we have substantially expanded experiment analyses as other reviewer recommended to further improve transparency and reproducibility.

---

### Official Review · Reviewer_wT3U · 2025-10-29

**Soundness:** 3
**Presentation:** 3
**Contribution:** 3
**Rating:** 4
**Confidence:** 5

**Summary:**

The paper proposes AURA (Affective Understanding via Robust Archetypes), a prompt-free framework for affective computing that operates in the frozen visual embedding space of CLIP. AURA unifies facial expression recognition (FER), action unit detection (AUD), and valence-arousal estimation (VAE) through a shared paradigm based on learnable visual archetypes. These archetypes are discovered via a self-organized mechanism that adaptively allocates more prototypes to affective states with higher visual or semantic complexity. Predictions are made by cosine matching between projected features and fixed archetypes, enabling efficient inference. The method is evaluated across six benchmarks and reports state-of-the-art results with low computational overhead.

**Strengths:**

1. A new framework is presented that integrates three different affective tasks into a unified architecture.
2. The model is lightweight and computationally efficient, with a low number of parameters and floating-point operations (FLOPs), making it suitable for practical use.
3. An adaptive prototype allocation mechanism helps determine the number of prototypes for each class or state, based on the data, reflecting the emotional complexity of the task.
4. Task-aware regularization techniques are introduced for both classification and regression tasks, including score-based attraction and repulsion for continuous labels.
5. Evaluation is conducted across multiple datasets and tasks, such as FER, AUD, and VAE, with a clear comparison to state-of-the-art (SOTA) methods.

**Weaknesses:**

1. The use of only the visual branch of CLIP for purely visual tasks is a reasonable design choice, but this approach is widely adopted in recent literature and does not represent a conceptual novelty. The core mechanism relies on learnable prototypes, a well-established technique in metric and few-shot learning.
2. While the introduction of the term "archetype" may create the impression of conceptual novelty, the main technical difference seems to be the adaptive density of prototypes based on label complexity - a useful but incremental extension.
3. Interpretability is primarily demonstrated through post-hoc archetype visualizations and error diagnosis, but it is unclear how this approach can explain individual model decisions like attention-based or saliency-based methods. The current form of interpretability is more useful for dataset curation rather than understanding per-sample model behavior.
4. While archetype visualizations highlight annotation errors, the paper does not analyze cases where the model’s prediction is wrong despite correct labels, which would better test the robustness of the archetype-based representation.
5. The framework is applied separately to each task, using task-specific archetypes and heads. This limits the claim of "unified modeling", as there is no shared representation or joint optimization across FER, AUD, and VAE.

**Questions:**

1. The authors could explain why they use the term "archetype" beyond stylistic or motivational reasons. Is there a specific reason why this term is necessary for their approach? Is there any formal distinction between their approach and other methods of adaptive prototype learning?
2. How would AURA explain a correctly labeled but misclassified sample?
3. It would be interesting to see if it is possible to train a shared set of archetypes for FER, AUD, and VAE using multi-task learning. If this is not possible, what are the fundamental reasons for this?

---

> ### Author Response · Authors · 2025-11-24
> **Author's Response - Thank you for your feedback (part 1)**
>
> We sincerely thank the reviewer for the thoughtful and valuable feedback. We appreciate the opportunity to improve the clarity and quality of our work. All modifications made in the manuscript and Appendix are highlighted in **Blue**. Our detailed responses to each comment are provided below.
>
> ---
>
> ### Question 1
>
> ---
>
> > Question 1. The authors could explain why they use the term "archetype" beyond stylistic or motivational reasons. Is there a specific reason why this term is necessary for their approach? Is there any formal distinction between their approach and other methods of adaptive prototype learning?
>
> **(1) Conceptual Motivation for the Term “Archetype”**
> - The term *“**archetype**”* in our work is not stylistic; it reflects a precise conceptual motivation. Originating from analytical psychology, **Carl Jung** described archetypes as **a primordial and recurring pattern that shapes perception and meaning** [1]. In contrast, the notion of a “**prototype**” in cognitive psychology refers to **the central or average member of a concept** [2]. Prototypes summarize categories by central tendency, whereas archetypes describe canonical structures that capture representative modes rather than averages.
> - We reinterpret archetypes geometrically: Unlike a prototype, which acting as global class means, archetypes correspond to local semantic extreme points that anchor distinct convex neighborhoods on the visual manifold. Whereas a prototype contracts the entire class distribution into one isotropic average point, and thus loses intra-class variability and interpretability, an archetype captures an **extremal, locally coherent semantic pattern** that recurs within a specific region of the manifold. By spanning multiple such extreme points, archetypes preserve fine-grained structure, model **smooth transitions** across semantic space, and provide more **interpretable variation** than prototype-based representations.
>
> **(2) Fundamental Distinction from Prototype Learning**
>
> AURA differs from standard prototype-learning methods [3, 4, 5] in interpretability, representational geometry, and modeling capability:
>
> - **(a) Interpretability — Archetypes as clear semantic anchors.**  Prototype methods compress an entire class into **one global average**, producing blurred, weakly interpretable representations. Archetypes instead form **multiple localized, visually coherent semantic anchors**, each capturing a distinct configuration (e.g., “high-valence smile,” “sadness with tightened eyelids”), offering far richer interpretability.
> - **(b) Representation — Archetypes preserve structured variability.**   Prototypes collapse all intra-class variability into **a single point**, discarding multi-modal structure. Archetypes **decompose** the semantic space into **multiple meaningful subregions**, capturing fine-grained differences, heterogeneous appearance patterns, and smooth semantic transitions that a single prototype cannot represent.
> - **(c) Capability — Archetypes naturally support continuous targets.**   Prototype learning assumes discrete classes and lacks a mechanism to express local variations along a **continuous** scale. Archetypes automatically populate the continuous range with **multiple local anchors**, enabling piecewise modeling of nuanced perceptual shifts—something fundamentally impossible with a single prototype.
> - **(d) Discovery — Archetypes adaptively determine the number of modes.** Traditional prototype learning often relies on **pre-defined fixed** centroids or clusters, forcing data to conform to external assumptions. AURA's Archetypes automatically and **adaptively determine the necessary number** of visual modes that emerge naturally from the feature space, ensuring optimal, data-driven representation without fixed priors.
>
> **(3) Why the Term “Archetype” Is Necessary**
> Based on the distinctions above, “archetype” is not interchangeable with “prototype” because:
> - AURA models **multiple coherent modes per class**, not a single representative.
> - AURA’s archetypes cover **continuous semantic ranges**, not discrete category centers.
> - AURA models **structured variability**, not global averages.
> - AURA yields **interpretable, recurring visual motifs**, aligning with Jung’s original meaning.
> - AURA uses **adaptive, context-aware assignment**, unlike static prototype selection.
>
>
> [1] Jung et. al. "The archetypes and the collective unconscious (Vol. 9)." *Trans. RFC Hull. Princeton* (1968).
> [2] Osherson et. al. "On the adequacy of prototype theory as a theory of concepts." Cognition (1981).
> [3] Snell et. al. "Prototypical networks for few-shot learning." *Advances in neural information processing systems* (2017).
> [4] Perner et. al. "Prototype-based classification." *Applied Intelligence* (2008)
> [5] Schlinge et. al. "Comprehensive evaluation of prototype neural networks." *arXiv preprint arXiv:2507.06819* (2025)..

---

> ### Author Response · Authors · 2025-11-24
> **Author's Response - Thank you for your feedback (part 2)**
>
> ### Weakness 1 & 2
>
> ---
>
> > Weakness 1. The use of only the visual branch of CLIP for purely visual tasks is a reasonable design choice, but this approach is widely adopted in recent literature and does not represent a conceptual novelty. The core mechanism relies on learnable prototypes, a well-established technique in metric and few-shot learning.
> >
> > Weakness 2. While the introduction of the term "archetype" may create the impression of conceptual novelty, the main technical difference seems to be the adaptive density of prototypes based on label complexity - a useful but incremental extension.
>
> Here we address both concerns and emphasize that AURA’s novelty does **not** come from using CLIP’s visual branch, nor from repackaging conventional “learnable prototypes.” Instead, AURA introduces a **fundamentally different matching paradigm**, a **fundamentally different representational target**, and a **backbone/task-agnostic mechanism** that clearly differentiates it from prior VLM or prototype-based approaches.
> - **(1) New Alignment Paradigm: Visual $\rightarrow$ Archetype — Beyond CLIP's Visual $\rightarrow$ Text**: The core conceptual innovation of our AURA is the shifts from the standard Visual $\rightarrow$ Text alignment to Visual $\rightarrow$ Archetype matching. This allows the model to align samples with automatically learned canonical visual modes emerging from the data, fundamentally changing how VLM feature spaces are used to discover structured semantic patterns without relying on text priors or prompt engineering.
> - **(2) New Representational Target: Sharp Semantic Anchors — Beyond Blurry Centroids**: As explained in Question 1, The core difference is representational, which is a **representational shift**, not a numerical refinement:
> 	- Prototypes model averages (blurry, low-semantic centroids);
> 	- Archetypes model multiple coherent canonical visual patterns (semantic articulation/interpretable visual anchors).
> 	Thus, AURA's archetypes preserve fine-grained semantic structure by capturing **distinct visual variants**, a capability fundamentally obscured by single-centroid prototype methods.
> - **(3) Mechanism Agnosticism: Task & Model-Agnostic — Beyond Discrete and Model-Limited**: AURA's mechanism operates entirely in feature space, making it uniquely both task and model-agnostic. Traditional prototype methods are inherently restrictive, often being model-limited (tightly coupled to specific architectures) and task-limited (assuming only discrete classes). In stark contrast, AURA's framework naturally extends to **continuous** regression tasks (e.g., valence/arousal) and achieves consistent improvements even when applied to **non-VLM backbones** like DINO. This versatility confirms that the strong performance gains originate purely from the powerful archetype mechanism itself.

---

> ### Author Response · Authors · 2025-11-24
> **Author's Response - Thank you for your feedback (part 3)**
>
> ### Weakness 3
>
> ---
>
> > Weakness 3. Interpretability is primarily demonstrated through post-hoc archetype visualizations and error diagnosis, but it is unclear how this approach can explain individual model decisions like attention-based or saliency-based methods. The current form of interpretability is more useful for dataset curation rather than understanding per-sample model behavior.
>
> It is important to clarify that AURA is designed to provide **semantic-level interpretability** rather than pixel-level attribution, and this design yields **complementary strengths** rather than limitations. Furthermore, if pixel-level attribution is desired, it remains **fully achievable** by jointly training the entire backbone along with the AURA module. Our detailed response is as follows:
>
>  - **(1) Pixel-level saliency/attention is not our objective — and is compatible**
> 	 - **AURA Targets Semantic Decisions, Not Pixel Gradients**: is important to clarify that AURA’s interpretability is **not** meant to replicate pixel-level saliency or attention maps, because our objective is fundamentally different. AURA is designed to explain **semantic decision structure**—how a model organizes high-level visual concepts—rather than pixel-level gradients. This is not a limitation but a **deliberate design choice** aligned with our goal of probing VLM semantic spaces.
> 	 - **Pixel-Level Attribution Is Compatible**. Importantly, AURA does **not** prevent pixel-level explanations. If AURA is **trained jointly** with the backbone (rather than on frozen features), all standard pixel-level interpretability methods can be applied immediately. This opens an exciting direction—in line with the reviewer’s suggestion—toward a **dual-level interpretability framework**: pixel-level attribution combined with semantic-level attribution to fully understand model's behavior.
> 	 - **AURA adds semantic-level interpretability beyond what CLIP/DINO currently offer**: While dozens of existing methods provide **pixel-level attribution** for popular pretrained models like CLIP and DINO, very few methods explain these models from a **visual-semantic perspective**. AURA offers semantic-level insights into **how these model representations are structured and how they align with high-level semantics**—a dimension of interpretability that saliency maps simply cannot provide.
>  - **(2) AURA explains per-sample model behavior via explicit semantic matching**
> 	 - AURA **can** explain per-sample model behavior at the **semantic** level. For every input, AURA makes its decision through a transparent process: it shows ***which*** archetype the sample aligns to, ***why*** that archetype is selected, and ***what*** canonical visual pattern that archetype represents.
> 	 - This provides a direct, concept-level explanation of the model’s belief about the sample (e.g., “this image matches the low-intensity AU12 archetype”), offering insight into ***how*** the model organizes high-level semantics—**a type of interpretability that saliency maps cannot provide**. Unlike attention-based methods that only reveal ***where*** the model focuses, AURA reveals **what** semantic pattern the model associates with the sample and ***why***, providing intrinsic, interpretable per-sample reasoning directly aligned with ***how*** VLMs operate in feature space.
> - **(3) Dataset-level diagnosis is a bonus, not a limitation—and it *comes from* per-sample model behavior..**
> 	- First of all, dataset-level diagnosis is a bonus that most models cannot provide.
> 	- Second, AURA’s dataset-level curation ability is *not* separate from per-sample behavior interpretability — it arises *precisely because* AURA explains model behavior at the sample level. Each sample’s archetype assignment directly shows (i) **how** the model interprets that sample, (ii) **which** semantic pattern it activates, and (iii) **whether** it deviates from expected modes. When samples activate an unexpected archetype, this naturally reveals mislabeled items, rare modes, or annotation inconsistencies. In other words, **dataset-level insights are the aggregation of per-sample semantic decisions**, not a limitation of the method. Instead, they demonstrate how precisely AURA captures **sample-level model behavior** in semantic space.

---

> ### Author Response · Authors · 2025-11-24
> **Author's Response - Thank you for your feedback (part 4)**
>
> ### Weakness 4 & Question 2
> -----------
>
> > Weakness 4. While archetype visualizations highlight annotation errors, the paper does not analyze cases where the model’s prediction is wrong despite correct labels, which would better test the robustness of the archetype-based representation.
> > Question 2. How would AURA explain a correctly labeled but misclassified sample?
>
> Analyzing failure cases is key to fully understanding AURA’s representational robustness. We have added a dedicated failure-case analysis in the revised manuscript, illustrated using **blue** highlights in **Appendix D (Figure 7)**. In these examples, we additionally demonstrate that AURA’s model behavior can be clearly explained through a **which–why–what–how** structure.
>
> As illustrated in Figure 7 (left top panel error sample with GT label of Happiness), even when the categorical prediction is incorrect, the assigned archetype remains **semantically meaningful** and provides a comprehensive model behavior analysis:
>
> * AURA shows **_Which_** Archetype it Aligns To: For the Happiness sample misclassified as Anger, AURA explicitly identifies the closest anchor—Anger P1—the archetype whose facial configuration the sample most resembles.
> * AURA explains **_Why_** the Match Occurs: The sample displays a "scream-like" pattern (wide-open mouth, lower-face tension) that is visually closer to Anger P1 than to any of its true label's Happiness archetypes.
> * AURA makes explicit **_What_** Visual Concept Defines the Archetype: By displaying the support images, AURA clarifies the precise visual concept that defines Anger P1—a shared, structurally consistent facial pattern.
> * AURA demonstrates **_How_** the Model Organizes Semantics: AURA reveals the structural basis of the error: the sample is placed in the region of semantic space populated by "scream-like" faces—an area where the dataset is **asymmetrically represented** across categories (frequent in Anger, rare in Happiness), thereby exposing a dataset-driven imbalance.
>
>
> This example shows that AURA does not merely report a wrong prediction; it provides a clear, per-sample semantic explanation of which archetype drives the decision, why the model prefers that archetype, what facial pattern defines it, and how the model’s semantic structure leads to this outcome. we demonstrate that AURA can:
>
> * **(i) Re-organize samples** based on **fine-grained visual semantics** (archetypal structure) rather than coarse-grained emotion categories (categorical labels), offering a more **precise semantic partition** of the expression space under weak supervision.;
> * **(ii) Reveal the semantic basis of the model’s mistake**, i.e., the sample is pulled toward an archetype whose **facial-muscle configuration is more similar than that of its label**; and
> * **(iii) Expose dataset-driven structural imbalances**, revealing situations where a specific facial configuration is underrepresented in its category but is overrepresented within the archetypes of a different emotion category. This clarifies **why** the model is structurally "pulled" to the wrong prediction.
>
> Taken together, these results address the reviewer’s concern by showing that AURA’s interpretability goes **far beyond detecting annotation errors**: it provides **robust, per-sample semantic explanations** for model failures and clarifies the structural factors that lead to them.

---

> ### Author Response · Authors · 2025-11-24
> **Author's Response - Thank you for your feedback (part 5)**
>
> ### Weakness 5 & Question 3
> -----------
>
>
>
> > Weakness 5. The framework is applied separately to each task, using task-specific archetypes and heads. This limits the claim of "unified modeling", as there is no shared representation or joint optimization across FER, AUD, and VAE.
> > Question 3. It would be interesting to see if it is possible to train a shared set of archetypes for FER, AUD, and VAE using multi-task learning. If this is not possible, what are the fundamental reasons for this?
>
> To prevent any misunderstanding, we wish to clarify that the "unified" aspect of AURA refers to a **unified paradigm**, not multi-task joint optimization across FER, AUD, and VAE in this specific study. We have used **blue** text （Line 82-84） in the revised manuscript to refine our description and avoid potential ambiguity. Below, we detail our intended meaning of "unified" and discuss the **exciting, yet challenging, direction** toward a fully multi-task framework:
>
>
> **(1) Clarification on the Meaning of “Unified” in Our Framework**
> - We do **not** intend to use the term “unified modeling” to suggest joint multi-task training across FER, AUD, and VAE. Instead, our use of the word *“unified”* refers to a **unified paradigm** or **unified task support framework**, which aims to provide a consistent modeling principle across diverse emotional analysis tasks. The phrase “Unified and Interpretable Modeling” in the contribution header is meant to describe this general paradigm rather than a single multi-task model. To avoid potential ambiguity, we have refined the wording in the revised manuscript.
> - Our intended meaning of “unified” is that AURA provides a **single, reusable, and task-agnostic framework** that supports a broad range of settings:
> 	- different **tasks** (binary detection, multi-class classification, continuous regression),
> 	- different **input structures** (patch-level features, global features),
> 	- different **backbones** (CLIP, DINO),
> 	- different **output spaces** (discrete labels, multi-hot labels, continuous VA values).
> - Naturally, each task requires its task-depended configuration, for instance, CE loss for classification and CCC loss for regression. These differences arise from task requirements, not from limitations of AURA itself. In fact, the ability to seamlessly support these diverse settings demonstrates **AURA’s flexibility and generality**.
>
> **(2) Exciting Directions: Toward a Unified Multi-Task Emotional Modeling Framework**
> - Regarding the reviewer’s suggestion (Question 3 and Weakness 5) about jointly modeling FER, AUD, and VAE in a single multi-task learning (MTL) framework, this is indeed an exciting direction, which is part of our future work. We strongly believe that combining AURA with a unified, interpretable multi-task emotional modeling framework represents an impactful and forward-looking direction for the community. However, several fundamental challenges must be addressed:
> 	1. **Co-annotation challenge.** Existing datasets are annotated independently for different tasks, with minimal overlap. Only a few datasets such as Aff-Wild2 contain combined labels, while others differ significantly in domain (in-the-wild vs. in-the-lab) and labeling protocol. Building a unified MTL framework ideally requires utilizing all existing datasets simultaneously, which is non-trivial.
> 	2. **MTL optimization challenges.** Current MTL methods suffer from issues such as negative transfer, gradient conflict, and task imbalance. Existing solutions (e.g., GradNorm, PCGrad, MDGA) operate on principles that are not directly compatible with AURA’s archetype-driven representation design. Effectively integrating AURA with MTL requires addressing these deeper structural mismatches.

---

### Official Review · Reviewer_GcZi · 2025-11-01

**Soundness:** 3
**Presentation:** 3
**Contribution:** 3
**Rating:** 6
**Confidence:** 3

**Summary:**

This paper introduces AURA, a prompt-free framework for affective understanding that operates directly on frozen CLIP visual embeddings. The core problem it addresses is the threefold limitation of existing VLM-based methods in affective computing: (i) incompatibility with continuous regression tasks (like Valence-Arousal), (ii) reliance on brittle and labor-intensive text prompts, and (iii) the lack of a unified paradigm across Facial Expression Recognition (FER), Action Unit Detection (AUD), and VA regression (VAE). AURA's main contribution is a "Self-organized Archetype Discovery" module, which learns a set of visual archetypes that serve as perceptual anchors.

**Strengths:**

1. The shift from a text-dependent, "text-image matching" paradigm to a "visual-archetype matching" one is a significant and compelling contribution. This prompt-free approach directly solves a major, practical bottleneck in applying VLMs to specialized domains like affective computing.

2. The ability of a single framework to successfully unify classification (FER), detection (AUD), and continuous regression (VAE) is a major strength. The paper's design choices, using global-level embeddings for FER/VAE and patch-level embeddings for AUD, combined with distinct regularization strategies, are well-motivated and empirically validated.

3. The paper achieves SOTA performance across all three task categories on six different benchmarks. It achieves this with the lowest parameter count and FLOPs compared to other SOTA methods. The inference-time design (omitting the contextualization module and using only a projector and cosine matching) makes it extremely practical for deployment.

**Weaknesses:**

1. While inference is exceptionally simple, the training process appears highly complex. The total loss is a weighted sum of three major components. $\mathcal{L}^{Arc}$ is itself a composite of three other losses 23, one of which ($\mathcal{L}^{Reg}$) has two entirely different formulations depending on the task. The paper provides little to no discussion on the sensitivity to the various weighting coefficients ($\lambda_{Proj}$, $\lambda_{Arc}$, $\lambda_{Contx}$) or the margins ($m$) used in the regularization losses. This complexity could be a significant barrier to reproducibility.

2. The paper claims to evaluate on a video benchmark (DISFA). However, the AURA architecture appears to be a frame-based processor. It uses patch-level features for AUD, but there is no mechanism described for modeling temporal dependencies across frames. Therefore, its excellent results on DISFA are for frame-level AU detection, not "video-level" analysis in a temporal sense.

**Questions:**

The "visual archetype" paradigm is very promising. Do the authors believe this approach could be generalized to other domains beyond affective computing, where VLMs also struggle with prompt-brittleness, such as fine-grained visual categorization (FGVC) or other visual regression tasks?

---

> ### Author Response · Authors · 2025-11-24
> **Author's Response - Thank you for your feedback (part 1)**
>
> We sincerely thank the reviewer for the thoughtful and valuable feedback. We appreciate the opportunity to improve the clarity and quality of our work. All modifications made in Appendix are highlighted in **Orange**. Our detailed responses to each comment are provided below.
>
>
> ### Weakness 1
>
> ---
>
> > Weakness 1. While inference is exceptionally simple, the training process appears highly complex. The total loss is a weighted sum of three major components. $\mathcal{L}\_{\text{Arc}}$  is itself a composite of three other losses, one of which ($\mathcal{L}\_\text{Reg}$) has two entirely different formulations depending on the task. The paper provides little to no discussion on the sensitivity to the various weighting coefficients ($\mathcal{L}\_\text{Proj}$, $\mathcal{L}\_\text{Arc}$, $\mathcal{L}\_\text{Contx}$) or the margins ($m$) used in the regularization losses. This complexity could be a significant barrier to reproducibility.
>
> We agree that training stability and reproducibility is an important concern, and it aligns with the core design principles of AURA. Although AURA includes several components, the overall training setup remains simple and reproducible. We have added **Appendix G**, where a dedicated analysis is provided (highlighted in **Orange**). To ensure full reproducibility, we will release all code, pretrained models, and training scripts upon acceptance. Below, we explain these parameters through both theoretical rationale and empirical evidence:
>
> **1. Sensitivity of the loss weights ($\lambda$).**
>
> In all our experiments,  The total loss $\mathcal{L}=\lambda{\text{Proj}}\mathcal{L}^{\text{VAS}}+\lambda\_{\text{Arc}}\mathcal{L}^{\text{Arc}}+\lambda\_{\text{Contx}}\mathcal{L}^{\text{Contx}}$ uses the setting $\lambda\_{\text{Proj}}=\lambda\_{\text{Arc}}=\lambda\_{\text{Contx}}=1$ without any tuning. This uniform configuration works consistently across all datasets (RAF-DB, EmotioNet, AffectNet) and all tasks (FER, AU, VA), indicating that AURA does not require delicate loss balancing.
>
> **1.1 Why equal weighting works**: The three losses govern *different representational layers* and therefore contribute gradients to ***disjoint parameter subsets***, their gradients do not interfere or compete in magnitude. : (i) $\mathcal{L}^{\text{VAS}}$ supervises the **projection head** and aligns archetype mixtures with task-level semantics. (ii) $\mathcal{L}^{\text{Arc}}$ shapes the **archetype structure** itself, controlling geometry, sparsity, and separation. (iii) $\mathcal{L}^{\text{Contx}}$ operates on the **contextual attention module** and adjusts cross-archetype message passing.
>
> **1.2 Balanced Structural within Archetype Regularization Loss $\mathcal{L}^{\text{Arc}}$.**
> - As for $\mathcal{L}^{\text{Arc}}$ mentioned by the reviewer, it shapes the archetype structure itself, controlling geometry, sparsity, and separation. More concretely, it consists of three components $\mathcal{L}^{\text{Arc}} = \mathcal{L}^{\text{Assign}} + \mathcal{L}^{\text{Dis}} + \mathcal{L}^{\text{Reg}}$, each serving a distinct structural purpose. Importantly, all three components operate on **normalized similarity measures** (e.g., cosine similarity and probability-simplex assignments), which naturally keeps their scales comparable.
> - These sub-losses push and pull archetypes in **complementary** directions ($\mathcal{L}^{\text{Assign}}$ promotes confident assignment, $\mathcal{L}^{\text{Dis}}$ enforces geometric separation, and $\mathcal{L}^{\text{Reg}}$ adds task-specific balancing forces) using equal weights ensures that no single force dominates the archetype geometry. Setting **all weights to 1** therefore maintains a stable equilibrium in the archetype space and avoids introducing artificial bias, leading to consistently stable training across all datasets.
> - Although $\mathcal{L}^{\text{Reg}}$ differs across tasks, all variants can be conceptually grouped into three structural constraints—*inter-class*, *intra-class*, and *diversity* regularization. Each of these sub-terms is intentionally designed to operate within a **similar magnitude**, ensuring that the archetype geometry is shaped by a balanced set of forces. These regularizers push archetypes apart when they become overly similar and pull them closer when they drift excessively far; as a result, any disproportionate increase in one component would introduce **directional bias**, such as collapsing archetypes or overspreading them. Maintaining these forces at comparable scales yields a stable equilibrium in the archetype space, explaining why $\mathcal{L}^{\text{Arc}}$ remains well-behaved under equal weighting.

---

> ### Author Response · Authors · 2025-11-24
> **Author's Response - Thank you for your feedback (part 2)**
>
> > Weakness 1. While inference is exceptionally simple, the training process appears highly complex. The total loss is a weighted sum of three major components. $\mathcal{L}\_{\text{Arc}}$  is itself a composite of three other losses, one of which ($\mathcal{L}\_\text{Reg}$) has two entirely different formulations depending on the task. The paper provides little to no discussion on the sensitivity to the various weighting coefficients ($\mathcal{L}\_\text{Proj}$, $\mathcal{L}\_\text{Arc}$, $\mathcal{L}\_\text{Contx}$) or the margins ($m$) used in the regularization losses. This complexity could be a significant barrier to reproducibility.
>
>
> **(continued authors' response)**
>
>
>
>
> **1.3 Experimental analysis**
>
> To examine the sensitivity of the loss-weight coefficients, we vary $\lambda \in {0.5, 1.0, 1.5}$. Across all settings and all datasets, the performance varies only minimally, with the main effect being slight changes in convergence speed.
>
>
> **Table1: Comprehensive sensitivity analysis of $\lambda$**
>
> | $(\lambda\_{\text{Proj}}, \lambda\_{\text{Arc}}, \lambda\_{\text{Contx}})$ | RAF-DB ACC (%) | Epoch | EmotioNet AU-F1 (%) | Epoch | AffectNet-VA CCC (%) | Epoch |
> | ----------------------------------------------------------------------- | -------------- | ----- | ------------------- | ----- | -------------------- | ----- |
> | $(1.0, 1.0, 1.0)$ **(default)**                                         | 94.0           | 152   | 67.3                | 136   | 74.1                 | 188   |
> | $(0.5, 1.0, 1.0)$                                                       | 93.8           | 187   | 67.1                | 165   | 74.1                 | 203   |
> | $(1.5, 1.0, 1.0)$                                                       | 94.0           | 149   | 66.8                | 138   | 74.2                 | 184   |
> | $(1.0, 0.5, 1.0)$                                                       | 94.2           | 176   | 67.1                | 148   | 73.9                 | 189   |
> | $(1.0, 1.5, 1.0)$                                                       | 94.3           | 142   | 67.2                | 136   | 74.0                 | 196   |
> | $(1.0, 1.0, 0.5)$                                                       | 93.8           | 180   | 66.9                | 164   | 74.0                 | 207   |
> | $(1.0, 1.0, 1.5)$                                                       | 94.1           | 146   | 67.2                | 122   | 74.2                 | 175   |
>
>
> **2. Sensitivity of the margin $m$**
>
> **2.1 Stability and Practical Range of the Margin $m$**
>
> **The margin $m$ serves the same purpose across tasks: it specifies the threshold that separates “should be close” from “should be separated.”**
> - For classification, the inter-class loss penalizes class centers whose cosine similarity exceeds $m$, meaning $m$ defines the maximum allowable similarity between different classes. Since cosine values lie in $[0,1]$, values in $m\in[0.2,0.4]$ provide a natural and stable range.
> - For regression, the attraction/repulsion losses use $m$ to distinguish pairs with similar vs. divergent predicted scores. Because valence/arousal targets span a broader semantic range, slightly smaller margins $m\in[0.1,0.3]$ work well.
>
>
> **2.2 Experimental analysis**
>
> To empirically validate the theoretical analysis, we perform a controlled sweep over a broad range of margin values. The results across all three tasks are reported in **Table 2**. As expected, performance only degrades when $m$ is set too small (leading to overly aggressive separation) or too large (resulting in insufficient discriminability).
>
> **Table2: Comprehensive sensitivity analysis of of $m$**
>
> | $m$       | Expression ACC (RAF-DB) | AU-F1 (EmotioNet) | VA-CCC (AffectNet-VA) |
> |-----------|--------------------------|--------------------|------------------------|
> | $m = 0.1$ | 93.2                     | 66.7               | 73.9                   |
> | $m = 0.2$ | 93.9                     | 67.2               | **74.1 (ours)**        |
> | $m = 0.3$ | **94.1 (ours)**          | **67.3 (ours)**    | 74.0                   |
> | $m = 0.4$ | 94.0                     | 67.3               | 73.8                   |
> | $m = 0.5$ | 93.6                     | 66.9               | 73.4                   |

---

> ### Author Response · Authors · 2025-11-24
> **Author's Response - Thank you for your feedback (part 3)**
>
> ### Weakness 2
>
> ---
>
> > Weakness 2. The paper claims to evaluate on a video benchmark (DISFA). However, the AURA architecture appears to be a frame-based processor. It uses patch-level features for AUD, but there is no mechanism described for modeling temporal dependencies across frames. Therefore, its excellent results on DISFA are for frame-level AU detection, not "video-level" analysis in a temporal sense.
>
> We adopt a frame-level setup solely for consistency with other tasks and to avoid redundancy, not due to any limitation. AURA is inherently task- and model-agnostic (validated by patch/global features and CLIP/DINO backbones), which can directly operate on **temporal features** by simply replacing the encoder. Also, even without temporal modeling AURA still surpasses recent temporal SOTA methods on DISFA.  Our clarification is summarized as follows:
>
> - **(1) AURA surpasses temporal SOTA methods even without temporal modeling**: Importantly, several recent DISFA methods (MDHRM, FUXI, AUNet in Table 1 main paper) explicitly incorporate  temporal modeling modules, yet AURA outperforms them using only single-frame inputs. This highlights the strength of the archetype formulation rather than any reliance on temporal cues.
> - **(2) AURA is encoder-agnostic and directly compatible with temporal features**: AURA operates purely in feature space; thus, replacing the backbone with any temporal encoder (e.g., TCN/GRU/Transformer-based video models) requires no modification. This encoder-agnostic nature has already been demonstrated using DINO in **Appendix F**, showing that AURA’s gains do not depend on CLIP’s properties. The same principle extends naturally to temporal encoders.
> - **(3) Frame-level AU detection is important for micro-expression analysis**: Subtle AU dynamics often occur within 40–100 ms (1–3 frames at 30 FPS). Hence,  treating video AU detection as a frame-level task is both standard and well aligned with how high-frequency facial dynamics are evaluated in practice.
> - **(4) Extending AURA to Temporal AURA is a promising future direction**: Beyond simply replacing frame features with temporal encodings, we will further extend AURA into a Temporal AURA, where archetypes are allowed to **evolve** over time to capture dynamic semantic patterns. This will enable modeling how canonical facial states transition, emerge, and fade across short temporal windows, offering a natural extension of AURA to video-based AU analysis.

---

> ### Author Response · Authors · 2025-11-24
> **Author's Response - Thank you for your feedback (part 4)**
>
> ### Question 1
>
>
> ---
>
> > Question 1: The "visual archetype" paradigm is very promising. Do the authors believe this approach could be generalized to other domains beyond affective computing, where VLMs also struggle with prompt-brittleness, such as fine-grained visual categorization (FGVC) or other visual regression tasks?
>
> Thank you for the encouraging comment and for highlighting the broader potential of the visual archetype paradigm. We agree that AURA is not restricted to affective computing and can generalize to a wide range of vision tasks.
>
> **(1) Generalization to fine-grained visual categorization (FGVC)**
> - AURA is naturally suited for FGVC because its archetype mechanism is explicitly designed to capture **subtle and structured intra-class variations**. As shown in *Figure 2* of the main paper, AURA discovers archetypes that correspond to **pose-dependent**, **texture-dependent**, and **part-dependent** visual modes. For instance, although *Sadness P1* and *Anger P1* both involve an open mouth, AURA distinguishes the expressions through subtle differences around the eyes and nose.
> - Since FGVC primarily relies on capturing **structural variations** (e.g., bird parts, vehicle components), rather than complex affective semantics, the same archetype-based representation aligns even more naturally with FGVC tasks.
>
> **(2) Applicability to general visual regression tasks**
> - AURA also extends naturally to continuous prediction problems such as age estimation, attribute scoring, or medical severity regression. Instead of collapsing the input to a scalar, AURA learns **multiple archetypes distributed along the continuous target range** (e.g., different regions of $[-1, 1]$ for VA). Each archetype represents a **local, interpretable anchor** on the semantic continuum, enabling structured modeling of smooth variations.
> - Our strong results on VA regression already show that this design handles continuous-valued targets effectively **without any architectural modification**, indicating natural applicability to broader regression tasks.
>
> **(3) Why AURA generalizes broadly: modality-agnostic, model-agnostic, and task-agnostic**
> - AURA’s components operate purely in **feature space**, making the method inherently independent of both the backbone architecture and the supervision format. This is supported by our DINO experiment (Reviewer 1, Table 2). More fundamentally, AURA generalizes well because its archetype mechanism is designed to capture **structured semantic variability**—a core challenge shared across diverse vision tasks, from fine-grained categorization to continuous regression and multi-appearance attribute prediction. Consequently, AURA is applicable to a wide spectrum of tasks where:
>     - **Multiple visual modes exist within the same label** (such as AU detection or fine-grained attribute recognition). AURA can represent each mode through distinct archetypes (e.g., pose-, texture-, or part-dependent patterns), ensuring robust modeling of heterogeneous appearances.
>     - **Fine-grained distinctions define class boundaries** (such as FER and FGVC). Since AURA specializes in separating subtle structural differences (as shown in Figure 2 for nuanced emotion cues), it directly transfers to domains where inter-class differences are small but visually consistent.
>     - **Outputs vary smoothly along a semantic continuum** (such as age estimation, valence–arousal regression, or medical severity scoring). AURA learns archetypes distributed along the continuous target range, allowing it to approximate a structured semantic manifold rather than mapping each sample to a single scalar.

---

### Official Review · Reviewer_ENGN · 2025-11-01

**Soundness:** 4
**Presentation:** 4
**Contribution:** 4
**Rating:** 6
**Confidence:** 2

**Summary:**

This paper presents AURA (Affective Understanding via Robust Archetypes), a visually interpretable framework for emotion understanding built on a frozen CLIP visual space. Instead of relying on textual prompts, AURA models emotions through adaptive visual archetypes that self-organize and contextualize within the embedding space, enabling unified handling of facial expression recognition, action unit detection, and valence-arousal regression. The approach combines efficiency and interpretability by matching features to archetypes through cosine similarity during inference, achieving competitive or superior results on six benchmarks with lower computational cost.

**Strengths:**

1. The paper proposes a clear and well-motivated framework that unifies multiple affective understanding tasks (FER, AU, VA) within a single, interpretable visual archetype space.
2. AURA achieves strong empirical performance across six benchmarks while maintaining low computational cost and providing visual interpretability through archetype-based reasoning.

**Weaknesses:**

The maximum number of archetypes ($K_{\max}$) is fixed (e.g., 98) without any accompanying sensitivity or stability analysis. This omission leaves uncertainty regarding how different choices of $K_{\max}$ affect model performance, convergence behavior, and the balance between representation granularity and computational efficiency.

**Questions:**

Since the paper is developed within the CLIP-based visual space, it would be interesting to further explore how AURA might behave when paired with a non-CLIP encoder. Such an investigation could offer additional insight into whether the framework’s strengths arise primarily from the archetype design or from the representational characteristics of CLIP features.

---

> ### Author Response · Authors · 2025-11-24
> **Author's Response - Thank you for your feedback (part 1)**
>
> We sincerely thank the Reviewer for the thoughtful and valuable feedback. We appreciate the opportunity to improve the clarity and quality of our work. All modifications made in the manuscript and Appendix are highlighted in **Green** and **Red**. Our detailed responses to each comment are provided below.
>
> ---
> ### Weakness 1
>
> ---
>
> > Weakness 1: The maximum number of archetypes ($K\_{max}$ ) is fixed (e.g., 98) without any accompanying sensitivity or stability analysis. This omission leaves uncertainty regarding how different choices of affect model performance, convergence behavior and the balance between representation granularity and computational efficiency.
>
> To improve clarity, we have added a comprehensive explanation in **Section 3** and **Appendix E** (highlighted in **Red**). Our clarification is summarized as follows:
>
> - To address the reviewer’s question, we first clarify the distinction between the **predefined upper bound** ($K\_{\max}$) and the **adaptively learned stable effective number** ($K\_{\text{stable}}$) of archetypes in AURA:
> 	- (1) $K\_{\max}$: a *predefined upper bound* (typically $150–400$), and
> 	- (2) $K\_{\text{stable}}$: a *stable number of active archetypes* that the model **automatically discovers during training** ($\sim 100$ for expression recognition on AffectNet/RAF-DB and $\sim 40$ for AU recognition on EmotioNet).
> 	Importantly, the value of ($K\_{\max}$) **does not determine** the final number of archetypes the model uses. We intentionally set ($K\_{\max}$) to be **over-complete** to ensure sufficient representational capacity, and then let the model adaptively determine how many archetypes are actually needed.
>
> - **The adaptivity is achieved by two key components**:
> 	- (1) Adaptive Archetype Regularization (Section 2.2.2), and
> 	- (2) Archetype Contextualization Module (Section 2.3).
> 	They jointly encourage useful archetypes to receive substantial assignments while suppressing redundant ones. During training, AURA determines active archetypes using a **concrete usage criterion (Appendix B)**. Let the archetypes codebook be $\mathcal{C}={e\_1,\dots,e\_{K\_{\max}}}$. During training, we track the global usage count $u\_k$ for each archetype, i.e., how often $e\_k$ is selected as the primal archetype. We compute its **normalized usage ratio** $\alpha\_k = \frac{u\_k}{\sum\_{j=1}^{K\_{\max}} u\_j}$. Archetypes with $\alpha\_k < \tau$ (e.g., $\tau = 0.01$) are considered under-utilized. In the early stage (first 20–30 epochs), any under-utilized archetype triggers the **Archetype Reset Mechanism (Appendix B)**, where its embedding $e\_k$ is reinitialized to prevent premature collapse. After this phase, the usage ratios stabilize, yielding a consistent $K\_{\text{stable}}$ (≈100 for FER, ≈40 for AU) regardless of the initial $K\_{\max}$. **At inference time**, archetypes with $\alpha\_k < \tau$ are pruned, resulting in a compact and semantically meaningful archetype dictionary.
>
> - **Sensitivity Analysis**: To directly address the reviewer’s concern, we have added a **sensitivity analysis w.r.t. ($K\_{\max}$)** (**Table 1** below). On RAF-DB, we vary $K\_{\max} \in {150, 200, 300, 400}$. As shown below, the learned ($K\_{\text{stable}}$) consistently lies around $99-102$, the accuracy varies within only $0.2$%, and the expression-wise archetype distribution remains highly similar. Convergence behavior (training epochs) is also comparable, without affecting the effective number of active archetypes, and although FLOPs increase with larger $K\_{\max}$, the cost remains low; AURA still operates with fewer FLOPs than the lightest SOTA model like MHAN (see Table 1 in main paper). These results demonstrate that AURA **automatically learns** an appropriate number and distribution of active archetypes, and is **robust and insensitive** to the specific initial choice of ($K\_{\max}$) in terms of model performance, convergence, and the granularity–efficiency trade-off.
>
> **Table 1. Sensitivity of AURA to the choice of ($K\_{\max}$) on RAF-DB.**
>
> | $K\_{\max}$ | Stable Archetypes ($K\_\text{stable}$) | ACC (%) | Convergence Epoch | Archetypes Distribution                                                      | Computational Efficiency （GFLOPs） |
> | ---------- | ------------------------- | ------- | --------- | ----------- | ------------ |
> | 150        | 100     | 94.1    | 153       | Anger 11 / Disgust 9 / Fear 6 / Happiness 28 / Neutral 19 / Sadness 12 / Surprise 15   | 0.26     |
> | 200        | 99  | 94.0    | 147    | Anger 11 / Disgust 9  / Fear 6 / Happiness 26 / Neutral 20 / Sadness 12 / Surprise 15    | 0.29                              |
> | 300      | 102      | 94.2    | 158               | Anger 11 / Disgust 10 / Fear 6  / Happiness 29 / Neutral 18 / Sadness 13 / Surprise 15   | 0.44                              |
> | 400        | 101             | 94.1    | 155      | Anger 12 / Disgust 10 / Fear 6  / Happiness 29 / Neutral 19 / Sadness 13 / Surprise 12   | 0.48            |

---

> ### Author Response · Authors · 2025-11-24
> **Author's Response - Thank you for your feedback (part 2)**
>
> ### Question 1
>
> ---
>
> > Since the paper is developed within the CLIP-based visual space, it would be interesting to further explore how AURA might behave when paired with a non-CLIP encoder. Such an investigation could offer additional insight into whether the framework’s strengths arise primarily from the archetype design or from the representational characteristics of CLIP features.
>
> In response to this comment, we extended our study to examine whether the improvements of AURA stem from CLIP’s pretrained feature space or from the archetype mechanism itself. To this end, we conducted additional experiments using **both CLIP [1] and DINO [2] encoders**, each evaluated under **two parallel configurations**: **(i) full encoder fine-tuning** and **(ii) frozen official encoder + AURA**. These new results are presented in **Appendix F** (highlighted in **Green**). The full experimental setup and findings are summarized below.
>
> - **CLIP-based Experiments**
> 	- **(i) CLIP Visual Encoder Full Fine-tuning (CLIP FT)**: We follow the standard CLIP training protocol and fully fine-tune the entire CLIP visual encoder together with the linear head for FER, AU detection, and VA regression—i.e., **all CLIP parameters are trainable**.
> 	- **(ii) AURA with Official Frozen CLIP Features (AURA-CLIP)**: We freeze the entire official pretrained CLIP visual encoder and use it strictly as a feature extractor, keeping **all backbone parameters fixed**. The _only_ trainable components are AURA’s archetype modules.
> - **DINO-based Experiments**
> 	- **(i) DINO Full Fine-tuning (DINO FT)**: Following the official DINO training recipe, we **fully fine-tune** the entire DINO student network, enabling all parameters to update for each task.
> 	- **(ii) AURA with Official Frozen DINO Features (AURA-DINO)**: We **freeze** the entire official pretrained DINO visual encoder and use it strictly as a feature extractor. Again, **only AURA’s archetype modules** are trainable.
>
> As shown in **Table 2 below**, AURA delivers **strong and consistent performance gains** under both the CLIP and DINO settings. Importantly, the relative improvements introduced by AURA remain stable across encoders and tasks. These results lead to three key conclusions:
> 1. **The gains of AURA are not dependent on CLIP’s pretrained alignment**, as comparable improvements are observed even when using fully self-supervised DINO features.
> 2. **AURA is encoder-agnostic**, consistently enhancing performance regardless of whether the backbone is CLIP or DINO.
> 3. **AURA effectively leverages the representational strengths of different pretrained encoders.** As expected, CLIP achieves higher overall accuracy than DINO due to its vision–language alignment and richer semantic priors. Notably, **the performance gains introduced by AURA are even larger on CLIP than on DINO**. This pattern shows that AURA can **fully exploit stronger, semantically aligned feature spaces**, resulting in more structured latent representations and superior interpretability and predictive performance.
>
>
> **Table 2. Performance of AURA with CLIP and DINO encoders. “FT” denotes full fine-tuning; “AURA” uses the official frozen encoder.**
>
> | Method        | Encoder                                   | RAF-DB ACC (%)  | AffectNet-VA CCC-VA (%) | EmotioNet AU-F1 (%) |
> | ------------- | ----------------------------------------- | --------------- | ----------------------- | ------------------- |
> | **CLIP FT**   | CLIP Visual Encoder (**finetuned**)       | 89.1            | 66.4                    | 62.3                |
> | **AURA-CLIP** | CLIP Visual Encoder (**official frozen**) | **94.0** (+4.9) | **74.1** (+7.7)         | **67.3** (+5.0)     |
> | **DINO FT**   | DINO (**finetuned**)                      | 88.3            | 65.4                    | 60.7                |
> | **AURA-DINO** | DINO (**official frozen**)                | **92.6** (+4.3) | **72.0** (+6.6)         | **65.4** (+4.7)     |
>
> [1] Radford, Alec, et al. "Learning transferable visual models from natural language supervision." _International conference on machine learning_. PmLR, 2021.
> [2] Caron, Mathilde, et al. "Emerging properties in self-supervised vision transformers." *Proceedings of the IEEE/CVF international conference on computer vision*. 2021.

---

### Author Response · Authors · 2025-12-02
**General response to all reviewers**

We sincerely thank all reviewers for their thoughtful feedback and positive recognition of our work. Reviewers highlighted AURA as  a ***significant and compelling shift*** from prompt-dependent text–image matching to a visual–archetype paradigm with a unified treatment of FER, AU, and VA (**GcZi**), a unified visual-archetype paradigm with ***originality*** and ***potential impact*** on affective computing and interpretable ML (**EDGc**),  a ***lightweight, efficient*** design with adaptive prototype allocation that reflects affective complexity (**wT3U**), and a ***clear and well-motivated framework*** with strong performance and efficiency (**ENGN**),

The main concerns raised by the reviewers are summarized and addressed as follows, for more detailed and reviewer-specific discussions, please refer to our point-by-point replies in the official review sections.

- **Archetype number & stability → We clarify the mechanism and add experimental evidence (ENGN, EDGc)**
    Reviewers asked about the sensitivity to the archetype upper bound and how the effective number is determined. We explicitly distinguish the predefined upper bound $K\_{\max}​$ from the learned effective number of active archetypes $K\_{\text{eff}}$, and describe the usage-ratio–based reset mechanism that drives self-organization. We also add a **systematic sensitivity study**, showing that performance, convergence, FLOPs, and per-class archetype distributions remain stable across a wide range of $K\_{\max}$​, confirming that AURA is robust to this choice.

- **Dependence on CLIP features → Our encoder-agnostic evidence (ENGN)**
  To clarify whether the gains come from CLIP or from AURA itself, we add **parallel experiments with CLIP and DINO** under both full fine-tuning and frozen-encoder + AURA settings. AURA brings consistent improvements in all configurations, showing that the benefits stem from the visual-archetype mechanism rather than CLIP alone.

- **Training complexity, hyperparameters, and temporal setup → Our robustness analysis (GcZi)**
  Reviewer GcZi requested more evidence on loss design and temporal modeling. We include a **dedicated sensitivity analysis** for the loss weights $\lambda$ and the margin $m$ across FER, AU, and VA, showing only minor performance variation and mainly changes in convergence speed. We further clarify that the main losses act on different parameter subsets and operate on normalized similarity measures, which justifies the simple equal-weight configuration.

- **Temporal modeling on DISFA → We clarify model-agnostic design and temporal extensibility (GcZi)**
    For DISFA, we clarify that our setup is frame-level AU detection for consistency with other tasks, and that AURA already **outperforms SOTA temporal methods** without any explicit temporal module. We further emphasize that AURA is **modeland encoder-agnostic**, operating purely in feature space, and can be **directly applied to temporal features** produced by video encoders (e.g., TCN/GRU/Transformer), making a “Temporal AURA” extension straightforward.

- **Archetype vs. Prototype → Archetypes as a psychologically grounded, semantically richer paradigm (wT3U)**
    Reviewer wT3U requested a clearer distinction between archetypes and prototypes. We ground our terminology in psychology and semantics: **prototypes** denote central-tendency category averages, whereas **archetypes** capture recurring canonical semantic patterns or local semantic extremes. AURA leverages archetypes to model multiple structured modes per label, support **continuous targets**, and **adaptively discover** the required number of modes. We also **strengthen interpretability** via per-sample analysis that shows **which** archetype is activated, **why** it is selected, and **how** this reflects dataset structure and model biases, providing semantically grounded explanations rather than only post-hoc visualizations.

Overall, we hope these clarifications and additional experiments adequately address the reviewers’ concerns and more clearly explain AURA’s core contribution and the scope of the unified visual-archetype paradigm.

---

### Meta-Review · Area_Chair_1YX9 · 2026-01-06

**Summary:**

This paper presents AURA, a visually interpretable framework for affective understanding that operates via adaptive visual archetypes in a frozen CLIP visual space, aiming to unify facial expression recognition, action unit detection, and valence-arousal regression in a single prompt-free paradigm. While reviewers acknowledged its originality, strong performance, efficiency, and potential impact, they raised several concerns: the initial lack of sensitivity analysis for the archetype upper bound, unclear dependency on CLIP features, high training complexity with insufficient hyperparameter justification, the absence of temporal modeling despite using video benchmarks, ambiguous distinction between "archetypes" and existing prototype methods, limited per-sample interpretability, and misleading claims about operating "directly" in CLIP space.

**Reviewer Concerns:**

In their rebuttal, the authors conducted new sensitivity experiments proving that AURA's performance is stable across a wide range of the archetype upper bound K_max. They demonstrated the framework's encoder-agnostic nature by showing consistent improvements with both CLIP and DINO backbones. They added comprehensive analyses on the sensitivity of loss weights and margins to address concerns about training complexity. They clarified the temporal modeling aspect on DISFA, explaining the frame-level evaluation and the straightforward extensibility to video encoders. Furthermore, they elaborated on the theoretical distinction between archetypes and prototypes, grounded in psychology, and enhanced interpretability by adding a failure-case analysis that explains misclassifications. Nevertheless, fundamental issues highlighted by the reviewers remain unresolved. The most significant is the overstatement of a unified model, as AURA uses separate, task-specific heads and was not trained jointly across FER, AU, and VA tasks. This means the claimed unification is more of a reusable paradigm than a truly integrated multi-task framework. Additionally, while the authors refined the terminology, the core technical novelty of the adaptive archetype mechanism is still perceived by some reviewers as an incremental extension of existing prototype-based learning methods rather than a paradigm shift. The initial misleading phrasing about operating directly in CLIP space, though corrected, points to a lingering concern about the precision of the methodological claims.

**Reviewer Scores:**

If the reviewers had been able to fully engage in a discussion based on the authors' rebuttal, their potential score adjustments would likely have been mixed and limited. Reviewers like ENGN and GCZI, who focused on empirical robustness, such as sensitivity to hyperparameters and encoder dependence, might have modestly raised their scores, perhaps from a 6 to 8, acknowledging that the new experiments directly addressed their methodological concerns. However, reviewers like wT3U and EDGC, whose critiques centered on fundamental conceptual issues, would almost certainly have maintained or even lowered their scores. They would argue that while the rebuttal added experiments, it did not resolve the core problem: the claimed "unified modeling" is a misnomer for a reusable paradigm with separate task heads, and the archetype concept, despite detailed distinction from prototypes, remains an incremental advancement rather than the claimed paradigm shift. Furthermore, the correction of the misleading "operates directly in CLIP space" claim might be seen as an admission of initial overstatement, reinforcing concerns about the paper's conceptual precision. Therefore, while the rebuttal strengthened the paper's empirical foundation, it did not sufficiently alter the fundamental novelty and contribution assessment for several reviewers, leading to an overall evaluation that would likely still fall below the acceptance threshold.

---

### Decision · Program_Chairs · 2026-01-26

Reject